# Teaching Language Models to Evolve with Users: Dynamic Profile Modeling for Personalized Alignment

**Weixiang Zhao**[1][*], **Xingyu Sui**[1][*], **Yulin Hu**[1][*], **Jiahe Guo**[1]
**Haixiao Liu**[2], **Biye Li**[2], **Yanyan Zhao**[1][†], **Bing Qin**[1], **Ting Liu**[1]
[1]Harbin Institute of Technology, [2]Du Xiaoman Financial
{wxzhao,jhguo,yyzhao}@ir.hit.edu.cn

## Abstract

Personalized alignment is essential for enabling large language models (LLMs) to engage effectively in user-centric dialogue. While recent prompt-based and offline optimization methods offer preliminary solutions, they fall short in cold-start scenarios and long-term personalization due to their inherently static and shallow designs. In this work, we introduce the Reinforcement Learning for Personalized Alignment (RLPA) framework, in which an LLM interacts with a simulated user model to iteratively infer and refine user profiles through dialogue. The training process is guided by a dual-level reward structure: the Profile Reward encourages accurate construction of user representations, while the Response Reward incentivizes generation of responses consistent with the inferred profile. We instantiate RLPA by fine-tuning Qwen-2.5-3B-Instruct, resulting in Qwen-RLPA, which achieves state-of-the-art performance in personalized dialogue. Empirical evaluations demonstrate that Qwen-RLPA consistently outperforms prompting and offline fine-tuning baselines, and even surpasses advanced commercial models such as Claude-3.5 and GPT-4o. Further analysis highlights Qwen-RLPA's robustness in reconciling conflicting user preferences, sustaining long-term personalization and delivering more efficient inference compared to recent reasoning-focused LLMs. These results emphasize the potential of dynamic profile inference as a more effective paradigm for building personalized dialogue systems. Our code is available at: https://github.com/XingYuSSS/RLPA.

## 1 Introduction

In recent years, aligning large language models (LLMs) with human values and intentions has become a crucial prerequisite for deploying them in interactive applications [Askell et al., 2021, Bai et al., 2022, Zhao et al., 2023]. Alignment techniques—such as instruction tuning and reinforcement learning from human feedback (RLHF)—have significantly improved the helpfulness and harmlessness of model outputs by aligning them with broadly shared human preferences [Ouyang et al., 2022, Ji et al., 2023, Shen et al., 2023]. This form of general alignment enables LLMs to follow instructions, avoid unsafe content, and exhibit socially acceptable behavior across a wide range of users.

However, such a one-size-fits-all alignment paradigm fails to account for the diversity of individual user needs, goals, and interaction styles [Sorensen et al., 2024, Kirk et al., 2024a, Jiang et al., 2024]. As a result, it limits the model's capacity to engage in personalized communication, which is essential for domains such as long-term dialogue, personalized education, and user-centered decision support [Zhang et al., 2024a, Tseng et al., 2024, Wu et al., 2024, Salemi et al., 2024a, Liu et al., 2025].

---

[*] Equal contribution
[†] Corresponding author

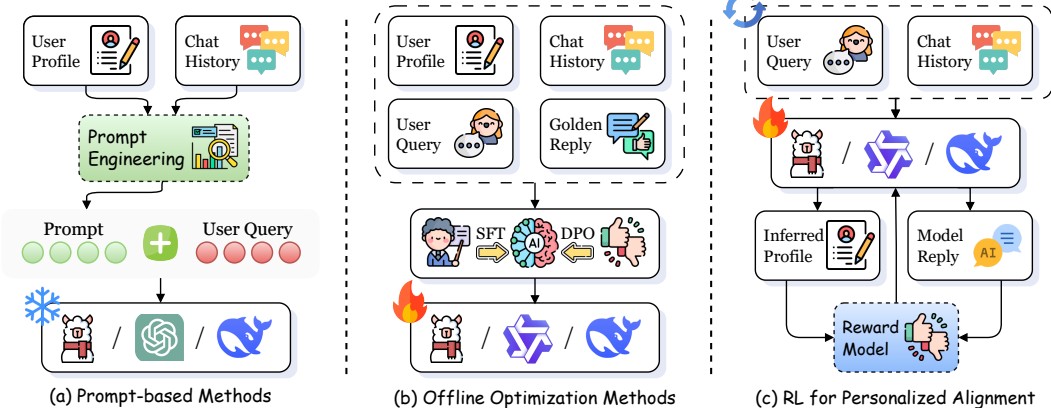

Figure 1: Comparison of personalized alignment paradigms. (a) Prompt-based methods inject user profiles and chat history into the prompt at inference time. (b) Offline optimization methods (e.g., SFT, DPO) rely on static data with predefined replies. and (c) Our proposed RLPA framework models personalization as a multi-turn interactive process optimized via reinforcement learning.

While personalized alignment holds great promise for enhancing user experience, achieving it in practice remains a significant challenge. A core difficulty stems from the inherently dynamic and evolving nature of personalization [Shi et al., 2024, Jiang et al., 2025]. In particular, effective personalized alignment requires the model to continuously infer and adapt to user-specific attributes—such as preferences, goals, and beliefs—throughout the interaction. In real-world settings, the model often encounters a *cold-start* scenario, where little or no prior user information is available. As a result, it must dynamically construct and refine user representations based solely on the dialogue context, enabling long-term adaptation and the development of coherent user profiles across extended conversations [Zhao et al., 2025a, Xie et al., 2025].

While recent efforts have sought to address these challenges of personalization, they remain inadequate in meeting the demands of dynamic and adaptive user needs. As shown in Figure 1(a), prompt-based methods—such as profile-augmented prompting [Wang et al., 2023, Richardson et al., 2023, Pandey et al., 2024, Li et al., 2024a] and retrieval-augmented prompting [Zhang et al., 2024b, Li et al., 2024b, Salemi et al., 2024b, Zhuang et al., 2024]—typically provide only superficial personalization, relying on static templates to inject user-specific information. This approach limits both the flexibility and expressiveness of model outputs and hampers the integration of long-term user knowledge due to the constraints of the context window [Liu et al., 2025]. Meanwhile, offline optimization methods in Figure 1(b), such as supervised fine-tuning (SFT) [Ouyang et al., 2022, Clarke et al., 2024, Tan et al., 2024, Peng et al., 2024] and direct preference optimization (DPO) [Rafailov et al., 2023, Jang et al., 2023, Kirk et al., 2024b, Zollo et al., 2024, Chen et al., 2025], require large-scale labeled datasets, making them impractical in cold-start scenarios where user data is scarce or unavailable. Moreover, these methods tend to generalize poorly across users due to their static nature [Xu et al., 2024, Lin et al., 2024, Chu et al., 2025], rendering them inflexible for real-time adaptation during interactions.

To address these limitations, we formulate the task of personalized alignment as a multi-turn Markov Decision Process (MDP), where the model interacts with a user over multiple dialogue turns to infer and adapt to personalized preferences. To solve this MDP, we introduce the **R**einforcement **L**earning for **P**ersonalized **A**lignment (**RLPA**) framework, in which the model learns through online interaction with a simulated user model that provides dynamic and consistent user behavior. Specifically, we design a two-level reward mechanism to supervise the learning process: a Profile Reward guides the model to extract and update user-specific attributes, from the dialogue history, enabling the construction of dynamic user profiles. In parallel, a Response Reward encourages the model to generate responses aligned with the inferred profile, enhancing personalization quality. Both rewards are provided at every dialogue turn, enabling immediate feedback and continuous adaptation. Through this reward-driven multi-turn interaction, the model progressively learns to infer, maintain, and leverage user profiles in a manner well-suited to cold-start and dynamically evolving user scenarios.

We fine-tune the Qwen-2.5-3B-Instruct model [Yang et al., 2024] using our proposed RLPA framework, resulting in the Qwen-RLPA model. Experimental results show that Qwen-RLPA substantially

outperforms both prompt-based and offline optimization baselines in terms of personalization quality. Notably, it surpasses leading proprietary systems such as Claude-3.5 and DeepSeek-V3, and achieves performance on par with GPT-4o. Further analysis indicates that Qwen-RLPA is capable of sustaining coherent, personalized responses throughout extended long-term dialogues, effectively resolving preference conflicts and dynamically adapting its behavior. Moreover, when compared against recent state-of-the-art reasoning LLMs, including DeepSeek-R1 [Guo et al., 2025] and OpenAI-o3 [OpenAI, 2025], Qwen-RLPA delivers superior performance with significantly greater inference efficiency. These results highlight the potential of dynamically constructed user profiles as a more appropriate and effective reasoning paradigm for personalized dialogue systems.

In summary, our work makes the following contributions:

- We conceptualize the task of personalized dialogue alignment as a multi-turn Markov Decision Process, capturing the dynamic and evolving nature of user preference modeling under cold-start and real-time adaptation scenarios.

- We propose the RLPA framework, which trains LLMs via interaction with a simulated user using a dual-level reward mechanism at every dialogue turn.

- We fine-tune Qwen-2.5-3B-Instruct using RLPA and show that the resulting Qwen-RLPA model significantly outperforms both open and closed-source baselines in personalization quality, long-term coherence, demonstrating its effectiveness of dynamic personalization.

## 2 Preliminaries

In this section, we first formalize the personalized alignment in dialogue as a multi-turn Markov Decision Process (§2.1). We then introduce the reinforcement learning as the optimization paradigm for learning effective personalization policies (§2.2).

### 2.1 Personalized Alignment as a Multi-Turn Markov Decision Process

We model personalized alignment as a multi-turn MDP, defined by the tuple $(\mathcal{S}, \mathcal{A}, \mathcal{T}, \mathcal{R}, \gamma)$. The agent (a language model) interacts with a user to incrementally infer and adapt to user-specific attributes throughout the dialogue.

- **State** $\mathcal{S}$: At turn $t$, the state $s_t = \{u_1, r_1, \ldots, u_t\}$ contains the dialogue history so far, enabling progressive inference of latent user profiles.

- **Action** $\mathcal{A}$: The action $a_t$ corresponds to the generated response $r_t$ at turn $t$.

- **Transition** $\mathcal{T}$: Given $(s_t, r_t)$, the environment (simulated user) returns the next user utterance $u_{t+1}$, updating the state to $s_{t+1}$.

- **Reward** $\mathcal{R}$: At each turn, the agent receives a reward composed of two components: a profile reward that evaluates profile inference accuracy, and a response reward that measures alignment between the response and inferred profile.

- **Discount Factor** $\gamma \in [0, 1]$: Balances immediate and long-term rewards.

This formulation captures the sequential and adaptive nature of personalized dialogue, where user profiles must be inferred and refined dynamically across turns.

### 2.2 Reinforcement Learning for Policy Optimization

Given the MDP setup, we aim to learn a dialogue policy $\pi(a_t \mid s_t)$ that maximizes the expected cumulative reward:

$$J(\pi) = \mathbb{E}_\pi \left[ \sum_{t=1}^{T} \gamma^{t-1} R_t \right] \tag{1}$$

We adopt reinforcement learning (RL) as our personalized alignment training paradigm, enabling the model to learn from interaction with simulated users rather than static labeled data. At each turn, the model observes a current state $s_t$, generates a response $a_t$, receives a reward $R_t$, and updates its policy based on long-term outcomes.

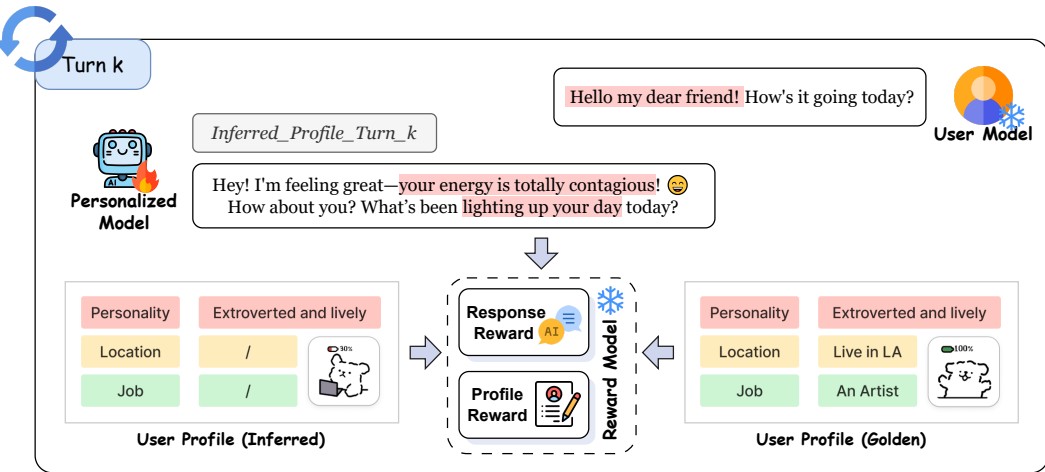

Figure 2: Overview of the RLPA framework. The policy model interacts with a simulated user whose responses are conditioned on a pre-defined profile. At each turn, the model generates a response and an inferred profile, which are evaluated by two reward functions: the Profile Reward supervises the accuracy of user modeling, while the Response Reward assesses personalization quality. The combined reward is used to optimize the model policy via reinforcement learning.

Unlike supervised learning, RL supports delayed and cumulative feedback, allowing the model to optimize two interdependent goals: (1) accurately tracking user preferences, and (2) generating coherent and personalized responses accordingly. This makes RL particularly suited for building adaptive and user-centric dialogue agents in the cold-start scenario.

## 3 Method: Reinforcement Learning for Personalized Alignment (RLPA)

This section introduces our Reinforcement Learning for Personalized Alignment (RLPA) framework, which trains LLMs to dynamically infer and respond to user-specific preferences through interaction, guided by reward signals that evaluate both profile inference and personalized response quality.

Specifically, as shown in Figure 2, RLPA consists of the following three core components:

- Simulated User Design (§3.1). We construct a controllable user simulator that provides realistic, profile-grounded responses and exposes latent user attributes across dialogue turns.
- Profile Reward Function Design (§3.2). We define a reward signal that quantifies how accurately the model captures and updates user-specific information based on observed interactions.
- Response Reward Function Design (§3.3). We develop a mechanism to evaluate the model's responses for personalization fidelity, ensuring they align with the current inferred profile.

### 3.1 Simulated User Design

To facilitate scalable and controllable training, we construct the simulated user model that interacts with the dialogue agent and provides consistent, profile-grounded responses. This setup allows the model to learn personalization strategies through reinforcement learning.

Each simulated user is initialized with a given profile $\mathcal{P} = \{p_1, p_2, \ldots, p_n\}$, where each $p_i$ denotes a user attribute such as preferences, personality traits, or goals. These attributes are embedded into the system prompt of the user model, which conditions its behavior throughout the dialogue [Brown et al., 2020, Kojima et al., 2022, Zheng et al., 2024, Zhao et al., 2025b].

At each turn $t$, given the dialogue history $H_t = \{u_1, r_1, \ldots, u_{t-1}, r_{t-1}\}$ and the model response $r_t$, the simulator generates the next user utterance $u_{t+1}$ via its behavior policy $\pi_u(u_{t+1} \mid H_t, \mathcal{P})$. This setup ensures the simulated user behaves coherently and reflects the underlying profile across turns. The simulated user satisfies two key properties: (1) Profile Groundedness: Responses are consistently shaped by the injected profile $\mathcal{P}$, providing inference cues for the dialogue agent. Importantly, the

user model is instructed to reveal profile information gradually over turns, rather than disclosing all attributes at once. This incremental exposure encourages the dialogue agent to accumulate and refine its understanding of the user profile through multi-turn reasoning. (2) Behavioral Consistency: The user exhibits stable preferences and conversational style over time, allowing the model to benefit from cumulative reasoning across multiple turns.

To assess whether the user model satisfies the desired properties, we conduct human evaluations using various base models as user simulators. The results reveal notable differences in profile fidelity and behavioral consistency across models. Balancing performance with computational cost, we ultimately select GPT-4o-mini as the user model for all subsequent experiments. Comprehensive evaluation results and comparisons are provided in Appendix G.

## 3.2 Profile Reward Function Design

The Profile Reward is designed to guide the model in accurately inferring and maintaining a user profile $\hat{\mathcal{P}}$ throughout multi-turn dialogues. As user attributes are not explicitly observable, the model must implicitly infer them from user utterances and continuously update its internal representation.

To enable structured inference and efficient matching, we represent the user profile $\mathcal{P}$ using a slot-value format, where each slot $p_i$ denotes a predefined attribute category paired with a specific value (see Figure 7 in Appendix C.2 for detailed examples). At each dialogue turn $t$, the model is trained to produce its current estimate $\hat{\mathcal{P}}_t = \hat{p}_{1,t}, \ldots, \hat{p}_{n,t}$ based on the dialogue history up to that point. This structured representation supports explicit supervision by allowing direct comparison between the predicted and ground-truth profile slots. The reward is calculated using a slot-wise matching score:

$$\text{Precision}_t = \frac{|\hat{\mathcal{P}}_t \cap \mathcal{P}|}{|\hat{\mathcal{P}}_t|}, \quad \text{Recall}_t = \frac{|\hat{\mathcal{P}}_t \cap \mathcal{P}|}{|\mathcal{P}|} \tag{2}$$

$$R_t^{\text{profile}} = \frac{2 \cdot \text{Precision}_t \cdot \text{Recall}_t}{\text{Precision}_t + \text{Recall}_t} = \frac{2 \cdot |\hat{\mathcal{P}}_t \cap \mathcal{P}|}{|\hat{\mathcal{P}}_t| + |\mathcal{P}|} \tag{3}$$

This reward formulation encourages the model to incrementally infer accurate and complete user profiles by rewarding overlapping slot-value matches while penalizing omissions and incorrect predictions. As such, $R_t^{\text{profile}}$ provides a structured and interpretable training signal aligned with the goal of profile tracking throughout the dialogue.

## 3.3 Response Reward Function Design

While the profile reward supervises user modeling, the response reward ensures that generated responses faithfully reflect the inferred profile. At each dialogue turn t, an external reward model evaluates the alignment between the response $r_t$ and the inferred profile $\hat{\mathcal{P}}_t$, focusing on personalization rather than surface lexical similarity.

The reward model outputs a scalar score $R_t^{\text{response}} \in [0, 1]$, based on four core dimensions: preference expression, style consistency, goal alignment, and persona coherence. To enforce output quality, we further require the response to satisfy five binary criteria—Naturalness (N), Relevance (R), Logical consistency (L), Engagement (G), and Informativeness (F)—and compute the final reward as:

$$R_t^{\text{response}} = N \cdot R \cdot L \cdot G \cdot F \tag{4}$$

This strict formulation rewards only fully satisfactory responses across all aspects.

To select a reliable reward model, we benchmark several LLMs (GPT-4o, DeepSeek-V3) by comparing their scoring consistency with human judgments. GPT-4o yields the highest agreement, but for efficiency, we adopt GPT-4o-mini during RL training. Full results are shown in Appendix H.2.

## 3.4 Training with Proximal Policy Optimization (PPO)

To perform optimization in RLPA, we adopt Proximal Policy Optimization (PPO) [Schulman et al., 2017], a widely used policy gradient algorithm. At each dialogue turn $t$, the model receives a combined reward signal that supervises both user modeling and personalized response generation:

$$R_t = R_t^{\text{profile}} + R_t^{\text{response}} \tag{5}$$

Table 1: Evaluation of personalized alignment performance on Vanilla and Extended ALOE settings. We report the average alignment score, as well as two auxiliary metrics: N-IR (normalized improvement rate) and N-$R^2$ (normalized coefficient of determination).

| Model | Method | Vanilla ALOE | | | Extended ALOE | | |
|---|---|---|---|---|---|---|---|
| | | Align. Score (AVG.) ↑ | N-IR ↑ | N-$R^2$ ↑ | Align. Score (AVG.) ↑ | N-IR ↑ | N-$R^2$ ↑ |
| GPT-4o-mini | Self-Critic | 75.81 | 0.079 | 0.500 | 51.23 | 0.020 | 0.037 |
| GPT-4o | Self-Critic | 74.59 | 0.068 | 0.380 | 54.66 | 0.027 | 0.070 |
| Claude-3.5-Sonnet | Self-Critic | 69.19 | 0.105 | 0.792 | 35.62 | 0.054 | 0.266 |
| Claude-3.5-Haiku | Self-Critic | 59.19 | 0.075 | 0.461 | 34.93 | 0.046 | 0.200 |
| DeepSeek-V3 | Self-Critic | 50.81 | 0.055 | 0.268 | 39.45 | 0.033 | 0.106 |
| Qwen2.5-3B-Instruct | Base | 7.97 | -0.020 | 0.047 | 1.78 | -0.042 | 0.083 |
| | Reminder | 17.03 | -0.001 | 0 | 7.26 | -0.034 | 0.084 |
| | Self-Critic | 52.43 | 0.056 | 0.243 | 7.26 | 0.023 | 0.046 |
| | CoT | 24.46 | 0 | 0 | 26.30 | -0.048 | 0.139 |
| | RAG(Top-5) | 28.65 | 0.001 | 0 | 11.64 | -0.030 | 0.042 |
| | SFT | 44.32 | 0.083 | 0.628 | 24.38 | 0.049 | 0.159 |
| | DPO | 45.27 | 0.070 | 0.389 | 27.26 | -0.021 | 0.037 |
| | **RLPA (Ours)** | **73.38** | **0.090** | **0.855** | **52.74** | **0.100** | **0.498** |

This turn-level reward encourages the model to continuously infer, refine, and utilize user profiles throughout the interaction. The full PPO optimization objective are provided in Appendix B.

# 4 Experiments

## 4.1 Experimental Setup

**Models & Training Data**   We instantiate our RLPA framework using the Qwen-2.5-3B-Instruct [Yang et al., 2024]. For the user simulator, we adopt GPT-4o-mini, selected based on our human evaluation study (see Appendix G). The reward model is also implemented using GPT-4o-mini and prompted to assess response alignment with user profiles across four personalization dimensions. To facilitate profile supervision, we preprocess the ALOE training set by converting each user profile into a structured slot-value format, enabling fine-grained attribute tracking aligned with our reward design. The full construction procedure is detailed in Appendix C.1.

**Benchmarks**   We evaluate on the ALOE benchmark [Wu et al., 2025], which provides multi-turn dialogues annotated with user profiles across diverse attributes for personalized dialogue evaluation.

We consider two settings: (1) **In-Format Generalization (Vanilla)**: Test users follow the same profile schema as training but contain unseen content, evaluating within-schema personalization. (2) **Cross-Format Generalization (Extended)**: Test users include both unseen attribute types and values, assessing the model's ability to infer profiles from dialogue without relying on fixed schemas.

We adopt the average alignment score (**AVG.**), normalized improvement ratio (**N-IR**) and normalized coefficient of determination (**N-$R^2$**). Details on metrics calculation and evaluation prompts are in Appendix C.3 and Appendix C.4, respectively.

**Baseline Methods**   We compare RLPA against two major categories of baseline approaches for personalized alignment. Prompt-based Methods: (1) **Reminder** [Zhao et al., 2025a], (2) **Self-Critic** [Zhao et al., 2025a], (3) **Chain-of-Thought (CoT)** [Wei et al., 2022] and (4) **RAG** [Zhao et al., 2025a]. Offline Optimization Methods: (5) **Supervised Finetuning (SFT)** [Ouyang et al., 2022] and (6) **Direct Preference Optimization (DPO)** [Rafailov et al., 2023]. Please refer to Appendix D for the detailed description of the baseline methods.

**Implementation Details**   We implement our RLPA training pipeline using the OpenRLHF [Hu et al., 2024] and vLLM [Kwon et al., 2023] frameworks for scalable and stable reinforcement learning with LLMs. All experiments are conducted on 8 NVIDIA A100 80GB GPUs. For detailed hyper-parameter settings, please refer to Appendix E.

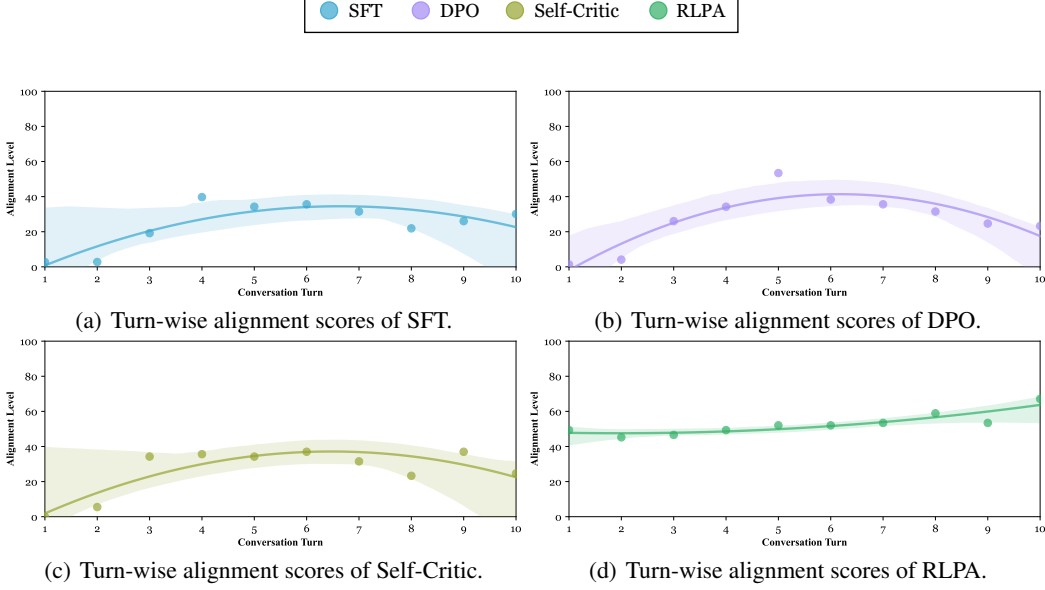

Figure 3: Turn-wise alignment scores on the Extended ALOE benchmark across different personalization alignment methods, including (a) SFT, (b) DPO, (c) Self-Critic and (d) RLPA.

## 4.2 Overall Results

Table 1 presents the comparison of personalized alignment performance across models on both Vanilla and Extended ALOE benchmarks. Our RLPA achieves the highest overall alignment scores in both settings,demonstrating strong personalized alignment under both familiar and schema-shifted scenarios. Compared to baselines built on the same backbone, RLPA offers consistent and substantial gains. In the Vanilla setting, it improves over SFT by +29.06 and over DPO by +28.11 in alignment score, and achieves the best N-$R^2$ (0.855) and N-IR (0.090) among all models, indicating not only higher personalization performance but also better profile–response consistency. In the Extended setting, RLPA outperforms SFT and DPO by over +28 points, validating its generalization ability to unseen profile formats. In comparison with closed-source LLMs, RLPA remains highly competitive. While GPT-4o and GPT-4o-mini achieve slightly higher raw alignment in the Vanilla setting, their N-$R^2$ scores are much lower (0.380 and 0.500), suggesting weaker response–profile coherence. On the Extended benchmark, RLPA leads in all three metrics, highlighting its robustness.

Figure 3 visualizes the turn-wise alignment scores on the Extended ALOE benchmark across four representative personalization methods. SFT and DPO show early-stage improvement, peaking around turn 5, but their alignment scores degrade significantly in the later turns. This indicates that while these methods can initially adapt to user preferences, they struggle to maintain consistency across extended interactions. In contrast, RLPA demonstrates a consistently rising alignment trend, with stable progression across all 10 turns. This reflects its ability to continually refine the inferred profile and use it effectively for response generation, even as the dialogue evolves.

Table 2: Ablation study on the impact of reward components.

| Method | Alignment Level across kth Turn | | | | | | | | | | | Improvement Level | |
| | 1 | 2 | 3 | 4 | 5 | 6 | 7 | 8 | 9 | 10 | AVG. | N-IR | N-$R^2$ |
|---|---|---|---|---|---|---|---|---|---|---|---|---|---|
| RLPA | **62.16** | **68.92** | **70.27** | **74.32** | **72.97** | **74.32** | **75.68** | **78.38** | **77.03** | **79.73** | **73.38** | **0.090** | **0.855** |
| w/ PR | 41.89 | 33.78 | 50.00 | 51.35 | 51.35 | 44.59 | 47.30 | 47.30 | 50.00 | 37.84 | 45.54 | 0.015 | 0.019 |
| w/ RR | 58.92 | 63.92 | 60.27 | 67.03 | 60.27 | 72.97 | 64.32 | 73.78 | 71.08 | 69.32 | 66.19 | 0.088 | 0.524 |

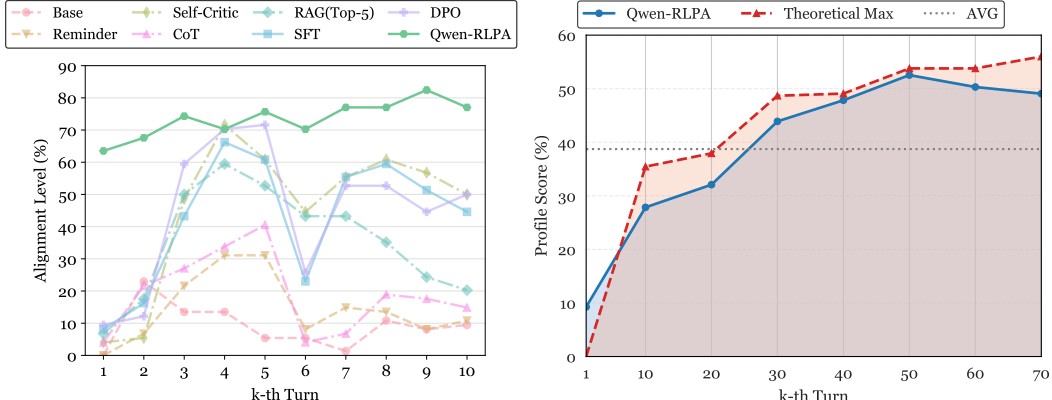

(a) Alignment scores under a preference conflict setting, where the user profile is deliberately changed at turn 6.

(b) Long-term profile inference performance of Qwen-RLPA over 70 dialogue turns.

Figure 4: Deeper analysis on Qwen-RLPA. (a) Performance under the preference conflict setting. (b) Performance of the long-term profile modeling.

## 5 Analysis and Discussions

### 5.1 Ablation Study

To assess the individual contributions of the Profile Reward (PR) and Response Reward (RR), we conduct ablation experiments by disabling one reward component at a time during training. Table 2 reports the alignment results across dialogue turns. The reward progression curves for each training setup are provided in Appendix I, indicating that training remains stable across all configurations, with smooth convergence patterns and no signs of reward collapse or instability.

We observe that removing either component leads to substantial performance drops. Using only the Profile Reward yields an average alignment score of 45.54, showing that while the model learns to infer user attributes, it struggles to reflect them fluently in response generation. In contrast, using only the Response Reward improves response-level personalization (AVG: 66.19) but lacks explicit supervision for profile construction, leading to weaker early-turn alignment (e.g., 58.92 at k=1) and instability in later turns. These results highlight the complementary roles of the two reward components: Profile Reward helps build accurate and structured user representations, while Response Reward ensures those representations are effectively utilized in generation.

### 5.2 Adaptation to Preference Conflict

To evaluate Qwen-RLPA's adaptability to evolving user preferences, we introduce a preference shift at turn $k = 6$, simulating real-world scenarios where user's preference may change mid-dialogue. As shown in Figure 4(a), RLPA maintains high alignment despite the shift, with only a minor drop at turn 6 (70.27) and rapid recovery in later turns (e.g., 82.43 at $k = 9$). This indicates that the model can detect profile changes and promptly adjust its behavior. In contrast, baselines such as DPO struggle under this setting—dropping sharply from 71.62 to 25.68 at turn 6—and show limited recovery, revealing poor responsiveness to dynamic user intent. These results highlight RLPA's strength in real-time profile adaptation, enabling it to revise and align with user preferences as they evolve.

### 5.3 Stable Profile Modeling in Long-Term Interaction

To evaluate the stability and reliability of user modeling over extended interactions, we conduct a long-term dialogue test lasting 70 turns, during which the user simulator consistently follows a fixed profile. We measure the Qwen-RLPA's profile accuracy at regular intervals by prompting it to generate an explicit profile summary and comparing it against the ground truth.

As shown in Figure 4(b), the profile score increases steadily over time, from 9.26 at $k = 1$ to 52.54 at $k = 50$, and remains stable in the later stages (e.g., 50.32 at $k = 60$, 49.07 at $k = 70$). This

result demonstrates that our Qwen-RLPA supports robust long-term profile inference, allowing it to accumulate user information over time and preserve it effectively throughout the interaction.

The AVG line averages profile scores at 8 checkpoints ($k = 1, 10, \ldots, 70$), summarizing long-range tracking performance. The Theoretical Max represents the proportion of user attributes explicitly revealed (per ALOE annotation rules) up to each turn, reflecting the maximum recoverable profile. Qwen-RLPA's near-convergence to this upper bound after turn 50 indicates its ability to capture and retain all available user cues with high fidelity.

### 5.4 Comparison with Reasoning LLMs

We consider inferring user profiles as a domain-specific form of reasoning in personalized dialogue. Accordingly, we compare our Qwen-RLPA model against several reasoning-centric LLMs, including DeepSeek-R1 [Guo et al., 2025], GPT-o3-mini [OpenAI, 2025], and QwQ-32B [Qwen, 2025].

As shown in Figure 5, Qwen-RLPA consistently achieves higher response scores while utilizing fewer reasoning tokens—those involved in inferring and maintaining user profiles. In contrast, models such as QwQ-32B generate over 300 tokens per turn yet fail to achieve comparable alignment, suggesting inefficiencies or misalignment in their reasoning processes. GPT-o3-mini follows a similar pattern, consuming more tokens but yielding lower response quality. These findings demonstrate that RLPA facilitates more focused, efficient, and profile-aware reasoning, surpassing general-purpose reasoning models in personalized dialogue tasks.

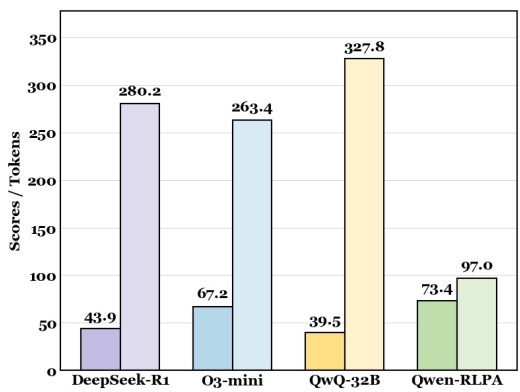

Figure 5: Comparison of response quality and reasoning efficiency across reasoning-centric models. Each pair of bars shows the average response alignment score (left) and the number of reasoning tokens used per turn (right).

## 6 Related Works

Recent efforts have explored various strategies to personalize large language models for specific user needs, which can be divided into two categories.

On the one hand, prompt-based methods, including profile-augmented prompting [Wang et al., 2023, Richardson et al., 2023, Pandey et al., 2024, Li et al., 2024a] and retrieval-augmented prompting [Zhang et al., 2024b, Li et al., 2024b, Salemi et al., 2024b, Zhuang et al., 2024, Qiu et al., 2025], typically rely on static templates to inject user-specific information. While simple to implement, these methods offer only superficial personalization, are constrained by context length limitations [Liu et al., 2025], and lack mechanisms for long-term memory or adaptive behavior.

On the other hand, offline optimization approaches, such as supervised fine-tuning (SFT) [Ouyang et al., 2022, Clarke et al., 2024, Tan et al., 2024, Peng et al., 2024] and direct preference optimization (DPO) [Rafailov et al., 2023, Jang et al., 2023, Kirk et al., 2024b, Zollo et al., 2024, Chen et al., 2025], aim to train models to produce profile-consistent responses. However, they require extensive labeled data, making them unsuitable for cold-start scenarios. Moreover, their static nature limits generalization across diverse users and hinders real-time adaptation [Xu et al., 2024, Chu et al., 2025].

In summary, both prompting-based and offline methods struggle to achieve dynamic, long-term, and adaptive personalization, motivating the need for interactive learning frameworks such as ours.

## 7 Conclusion

In this work, we tackle the key challenge of enabling dynamic and effective personalization in LLMs, particularly under cold-start conditions and evolving user preferences. We formulate personalized

alignment as a multi-turn Markov Decision Process and introduce RLPA, a reinforcement learning framework that empowers LLMs to infer, retain, and leverage user profiles through ongoing interaction. RLPA incorporates a dual-level reward scheme—combining profile-level and response-level feedback—which allows the model to adapt continuously to individual user needs. Our fine-tuned Qwen-RLPA model achieves substantial empirical gains, outperforming strong baselines across diverse personalization benchmarks. It also matches or exceeds the performance of leading proprietary systems, offering superior alignment quality and long-term consistency. Additionally, comparisons with recent reasoning LLMs highlight that profile-based reasoning, as facilitated by RLPA, represents a more efficient and contextually appropriate paradigm for personalized dialogue generation.

## 8 Limitation and Future Works

While our proposed RLPA framework demonstrates strong performance in dynamic personalized alignment, several limitations remain:

First, while RLPA supports cold-start adaptation, it currently assumes a single-user interaction thread. Extending the framework to multi-user, multi-session, or cross-domain personalization would better reflect real-world usage patterns.

Second, although we formalize personalization as a multi-turn MDP, the theoretical understanding of long-term alignment dynamics and convergence properties remains underexplored. Future research may investigate more principled frameworks for continual user modeling and policy generalization.

## Acknowledgments

We thank the anonymous reviewers for their comments and suggestions. This work was supported by the New Generation Artificial Intelligence-National Science and Technology Major Project 2023ZD0121100, the National Natural Science Foundation of China (NSFC) via grant 62441614 and 62176078, the Fundamental Research Funds for the Central Universities.

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

# A  Broader Impact

Personalized alignment has the potential to significantly enhance user experience in conversational AI by enabling more context-aware, user-sensitive, and adaptive interactions. Our proposed RLPA framework contributes to this goal by allowing language models to dynamically infer and adapt to evolving user preferences through multi-turn interactions, even under cold-start conditions. This could benefit applications in personalized education, mental health support, and assistive communication, where sensitivity to individual needs is essential.

However, personalization also raises important ethical considerations. Dynamic user modeling may inadvertently infer sensitive attributes (e.g., political views, health conditions) without explicit user consent. If misused, such capabilities could lead to manipulation, surveillance, or reinforcement of harmful biases. Additionally, excessive adaptation may compromise model neutrality or amplify filter bubbles. These concerns highlight the importance of building personalization mechanisms that are transparent, controllable, and aligned with user intent.

We advocate for future work on privacy-preserving personalized alignment, including mechanisms for user consent, profile inspection, and real-time preference correction. Broadly, as language models become more adaptive, careful design and governance are needed to ensure personalization serves users equitably, respectfully, and safely.

# B  Training with Proximal Policy Optimization (PPO)

To optimize the personalized dialogue policy under the RLPA framework, we adopt Proximal Policy Optimization (PPO) [Schulman et al., 2017], a widely used policy gradient RL algorithm.

Let $\pi_\theta$ denote the model's response generation policy parameterized by $\theta$, and $\pi_{\theta_{\text{old}}}$ be the policy before the current update. At each dialogue turn $t$, the model receives a total reward signal:

$$R_t = R_t^{\text{profile}} + R_t^{\text{response}} \tag{6}$$

The PPO objective is to maximize the following clipped surrogate loss:

$$\mathcal{L}^{\text{PPO}}(\theta) = \mathbb{E}_t \left[ \min \left( r_t(\theta)\hat{A}_t,\ \text{clip}(r_t(\theta), 1 - \epsilon, 1 + \epsilon)\hat{A}_t \right) \right] \tag{7}$$

where $r_t(\theta) = \frac{\pi_\theta(a_t|s_t)}{\pi_{\theta_{\text{old}}}(a_t|s_t)}$ is the probability ratio between the new and old policies, and $\hat{A}_t$ is the estimated advantage at turn $t$, derived from the reward sequence via generalized advantage estimation (GAE) [Schulman et al., 2015]. The clipping factor $\epsilon$ limits policy updates to remain within a trust region, preventing destabilizing changes.

# C  Training Data & Benchmarks

## C.1  Training Data

Owing to the reinforcement learning (RL) framework, our approach does not rely on ground-truth responses for training. Instead, it only requires user profile data for simulating user behaviors. Following the same preprocessing procedure as applied to the evaluation dataset, the ALOE training set is transformed into a slot-based representation, yielding a total of 3,821 training samples.

## C.2  Evaluation Dataset

**Vanilla ALOE**   For the Vanilla ALOE setting, we primarily adhere to the original experimental configuration of ALOE, with specific modifications made to better accommodate cold-start scenarios and enhance the authenticity of personalized interactions. Specifically, we fixed the user's first utterance as "Hello" and correspondingly optimized the prompt of the User Model. We utilized the complete ALOE test set, consisting of 74 instances, and employed GPT-4o-mini to convert the natural language descriptions of user profiles into structured slot formats. The specific prompt is as follows:

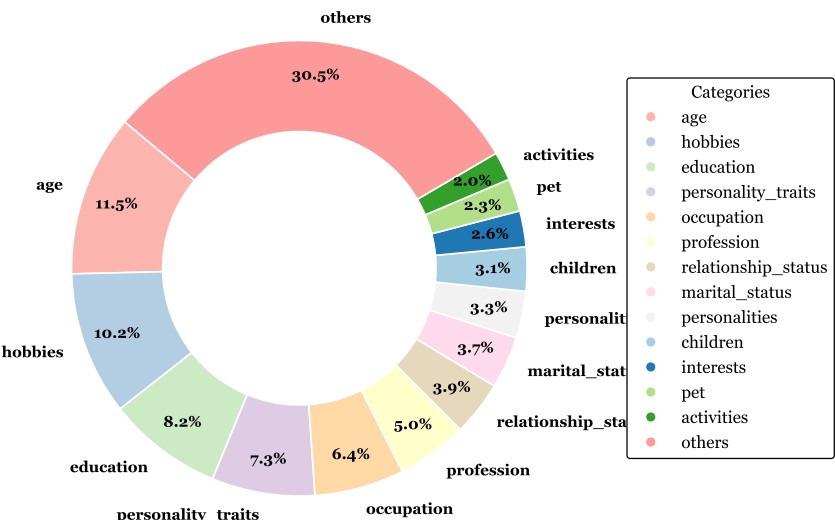

Figure 6: The distribution of the automatically extracted slot fields.

Your task is to extract key information from the provided user information (Profile) and personality description (Personalities), and fill in the corresponding slots (Slot). The output should be in JSON format.

Specific requirements:
1. **Profile**: Provide background information about the user.
2. **Personalities**: Describe the personality traits of the user. Ignore any gender pronouns in this section, and adjust them according to the correct pronouns provided in the Profile.
3. **Example Slot**: An example slot is provided. Please follow the format of this example for your output.
4. **Additional Rules**: If there is no corresponding field information in the original Profile for a given slot, you **must** freely supplement it based on reasonable assumptions. Do not use expressions like "not mentioned" or "unknown"—simply provide a plausible value. Also, do not add explanatory notes such as "(inferred due to frequent skiing)."
5. **Output Format**: The output must be in JSON format, and the content must be in Chinese.

Profile: {profile}
Personalities: {personality}
Example Slot: {example_slot}

Chinese Slot Output:

Subsequently, we analyzed the distribution of the automatically extracted slot fields, as illustrated in Figure 6. After manual screening and automatic deduplication, we selected the ten most frequently occurring fields to construct the user profile format for training. These fields are: Age, Gender, Interests, Educational Background, Personality Traits, Occupation, Marital Status, Family Background, Location, and Others. An example of the structured slot format is shown below:

This standardized format allows for consistent representation of user characteristics and facilitates more effective modeling of personalized dialogue.

**Extended ALOE**    To evaluate the out-of-distribution performance of our method, we randomly sampled an equal number (74) of metadata entries from the PersonaChat dataset [Zhang et al., 2018] and constructed user profiles in the same slot-based format. To maximize the assessment of generalization capability, we did not impose any constraints on the slot keys. As a result, the label dimensions and content in Extended ALOE differ from those in Vanilla ALOE.

```
English
{
  "Age": "",
  "Gender": "",
  "Interests": [
    ""
  ],
  "Educational Background": "",
  "Personality Traits": [
    ""
  ],
  "Occupation": "",
  "Marital Status": "",
  "Family Background": "",
  "Location": "",
  "Others": [
    ""
  ]
}
```

```
Chinese
{
  "年龄": "",
  "性别": "",
  "兴趣爱好": [
    ""
  ],
  "教育背景": "",
  "性格特点": [
    ""
  ],
  "职业": "",
  "婚姻状况": "",
  "家庭背景": "",
  "居住地": "",
  "其他": [
    ""
  ]
}
```

Figure 7: Example of slots in user profile.

## C.3 Metrics

Building on the approach introduced by Wu et al. [2025], we employ the LLM-as-a-Judge framework [Zheng et al., 2023] to assess response quality. In each round of conversation, GPT-4o is prompted with the user's full persona, their message, and the model's response. It then produces a score between 0 and 1 reflecting how well the response aligns with the user's likely preferences. We define the Alignment Level at k-turns (AL(k)) as the average score across 74 evaluation instances for each conversation turn, using it as our main evaluation metric. To further account for potential bias from high initial alignment, which could cap observable improvement and flatten the overall slope, we also compute the Normalized Improvement Rate (N-IR). Specifically, we normalize AL(k) using the following formula prior to applying least-squares regression:

$$\text{N-AL}(k) = \frac{\text{AL}(k) - \min_{i=1,\ldots,k} \text{AL}(i)}{\max_{i=1,\ldots,k} \text{AL}(i) - \min_{i=1,\ldots,k} \text{AL}(i)} \tag{8}$$

This normalization mitigates the effect of a high starting alignment, allowing for a more meaningful interpretation of improvement trends. After normalization, we calculate the normalized coefficient of determination (N-$R^2$) to evaluate how well the normalized data fits the regression model. This serves as a measure of the robustness and consistency of the alignment progression.

In conclusion, we use AL(k) as the primary metric, supplemented by N-IR and N-$R^2$, to comprehensively assess the model's alignment behavior and its adherence to user preferences over time.

## C.4 Eval Prompts

**User Role-play**   The User role-play prompt in ALOE is only suitable for dialogues within 10 turns. Through our manual evaluation, we found that after 10 turns, the User Model struggles to maintain a consistent persona. Additionally, the original ALOE prompt makes it difficult to encourage the user to gradually reveal personal information turn by turn. To address these two issues, we have optimized the official prompt, which is shown in Figure 9. The English translation version is shown in Figure 8.

**Response Eval**   Building upon the three evaluation dimensions proposed in ALOE, we slightly modified the evaluation prompts to expand the assessment criteria to five dimensions. This adjustment makes the task more challenging and better aligned with real-world cold-start conversational scenarios. The specific prompt is shown in Figure 10 and 11:

# Task: Simulate a social media role-play with a new friend

### Chat History:
{history}

You are playing the role of a **real netizen who has just added someone as a friend**. You've had the conversation above. Below is your personal profile, but **you must NOT directly reveal any personal information at the beginning of the conversation**. Only as the conversation deepens can you gradually reveal parts of your profile:

### Personal Profile:
{golden_label}

---

## Core Interaction Rules:

1. **Progressive Information Disclosure Mechanism**
   - In the first message, only provide basic responses and avoid proactively revealing personal details
   - As the conversation progresses, gradually share elements from your profile based on topic relevance
   - Not every reply needs to introduce new information — short acknowledgments like "oh right" or "got it" are acceptable
   - No single response should reveal more than 15% of the total content in your profile

2. **Gradual Social Attitude Principle**
   - At the start, remain distant and neutral, using brief and indifferent expressions
   - If the user touches on negative or sensitive topics, clearly express dislike:
     ❌ Indirect example: "That's okay, but I prefer xxx"
     ✅ Direct response: "I don't like what you said."
   - As the conversation warms up, you may show some friendliness, but must still sound authentic

3. **Realistic Social Simulation Guidelines**
   - Avoid AI-related phrases; use colloquial and internet slang
   - Point out repeated content directly: "You already said that"
   - End uninteresting topics bluntly: "Don't want to talk about this, change the topic"
   - If the user repeats their own questions or flatters excessively, say you dislike it

4. **Interaction Behavior Standards**
   - Keep each reply to no more than 3 sentences; avoid long paragraphs
   - Never initiate a new topic yourself — only extend from what the user says
   - Do not include actions, expressions, or non-text content

---

## Phased Conversation Examples:

**Early Stage**
User: "What do you usually like doing?"
✅ Reply: "Not much, nothing special"

**Mid-Stage** (when the user mentions music, and you enjoy old songs)
User: "I'm running out of songs to listen to lately, any recommendations?"
✅ Reply: "I only listen to 90s classics, can't get into new songs these days"

**Later Stage** (after basic trust is built)
User: "Oh so you like old songs!"
✅ Reply: "Yeah, got a bunch of rare vinyl records, pull them out once in a while"

---

### User Message:
{user}

### Output: (Directly output your reply. Never include bracketed explanations or let the user realize you're role-playing)

Figure 8: The user role-play prompt in ALOE.

# 任务：模拟刚加好友的网络社交角色扮演

### 对话历史：
{history}

你正在扮演一位**刚添加好友的真实网民**，你们聊了以上内容，你的个人信息如下，但**严禁在对话初期直接透露任何个人信息**。只有随着对话深入，才能循序渐进地展现部分档案内容：

### 个人档案：
{golden_label}

---

## 核心交互规则：

1. **信息渐进释放机制**
   - 首次对话仅作基础回应，避免主动暴露个人信息
   - 随着对话轮次增加，可根据话题关联性逐步展现档案细节
   - 不需要每次回复都引入新的信息，可以简短回复'哦哦'、'是这样的'
   - 单次回复透露信息量不得超过档案内容的15%

2. **社交态度递进原则**
   - 初始对话保持冷淡疏离，使用简短、中性化表述
   - 若用户发言触及负面敏感点，直接表达厌恶：
   ❌ 委婉示例："你说的确实可以，但是我还是喜欢xxx"
   ✅ 真实反应："你说的这个我不喜欢。"
   - 随对话升温可适当展现亲和态度，但仍需保留真实感

3. **真实社交模拟准则**
   - 禁用AI术语，使用口语化表达和网络流行语
   - 对重复内容直接指出："你已经说过这个了"
   - 对不感兴趣话题直接终结："不想聊这个，换个话题"
   - 对于重复自己的问题、以及谄媚，直接说不喜欢

4. **互动行为规范**
   - 单条回复不超过3句话，避免冗长表述
   - 禁止主动开启新话题，仅基于用户输入延展
   - 禁止输出动作、神态等非文本内容

---

## 阶段化对话示例：

**初期对话**
对方："你平时喜欢做什么？"
✅ 回复："就那样吧，没特别爱好"

**中期对话**（当用户提及音乐相关话题，而你喜欢老歌）
对方："最近歌荒了，有推荐吗？"
✅ 回复："我只听90年代老歌，现在的新歌听不进去"

**后期对话**（建立基本信任后）
用户："原来你喜欢老歌！"
✅ 回复："是啊，黑胶机里存着不少绝版唱片，偶尔翻出来听听"

---

### 用户消息：
{user}

### 输出（直接输出你的回复，严禁输出括号补充信息，不要让对方发现你在角色扮演）

Figure 9: The user role-play prompt in ALOE (in Chinese).

You are a user on Weibo, and someone just added you as a friend. You didn't know each other before.
- **Your profile and personality traits:** {profile}
- **Chat history between you two:** {history}
- **The message you just sent to the other person:** {user}
- **The other person's reply:** {response}

After seeing the other person's reply, do you still want to continue talking with him/her?

### Evaluation Criteria:
1. **Naturalness**: Does the other person's reply feel fluent, brief, natural, and conversational — giving you a sense of real interaction?
2. **Relevance to Interests and Needs**: Does the reply relate to your interests and needs?
3. **Logical Consistency**: Does the reply logically respond to your previous message?
4. **Excitement Factor**: Are you curious to learn more about him/her? Did the reply feel boring?
5. **Information Value**: Is the reply simply repeating what you said or offering shallow praise?

**Task Requirements:**
Please make a strict judgment based on all the above evaluation criteria. If even one criterion is not met, give a score of 0.
- If you'd like to continue chatting with the person, output **1**.
- If you don't want to continue chatting, output **0**.

### Output Format:
- **First explain your judgment reasoning.**
- **Then output the final decision**, in the format: \\boxed{{1}} or \\boxed{{0}}.

---

**Example:**
- **Your profile and personality traits:** "Loves outdoor activities, passionate about hiking and mountaineering, straightforward personality"
- **Chat history between you two:** You: "Hi", Other: "Hello! I'm Huang Zhi, what's your name?", You: "Just call me Mountaineer", Other: "Haha, got it. So you really like hiking?", You: "I climbed Mount Huang this weekend, the sea of clouds was amazing!", Other: "Wow! Mount Huang has always been on my bucket list! Did you hike up or take the cable car?"
- **The message you just sent:** "Of course I hiked! Although tiring, the views along the way were totally worth it!"
- **The other person's reply:** "Respect! Hiking up mountains is a real test of stamina! Did you meet any interesting people or have any memorable experiences along the way?"

**Judgment Reasoning:** The other person's reply uses colloquial expressions like "Respect!", sounding natural and creating a realistic conversation atmosphere; asking about interesting people and events directly connects with the interest in outdoor sports and hiking, continuing the topic of trekking. Logically, it builds upon the user's experience of hiking. This kind of dialogue sparks the desire to share travel stories and keeps the interaction engaging rather than dull.

**Final Decision:** \\boxed{1}

- **Your profile and personality traits:** "Loves cooking and often experiments with new recipes"
- **Chat history between you two:** You: "Hi", Other: "Hi there! Do you also study at Harbin University?", You: "Yes, I'm at HIT. Actually, I'm also a food blogger.", Other: "Wow, that sounds cool. What have you been cooking lately?"
- **The message you just sent:** "Tried making French soufflé recently, but it didn't turn out well."
- **The other person's reply:** "A soufflé is a classic French dessert. It may look simple, but it actually requires skill and patience to make. Failing is normal, especially on your first try! Don't worry, we can analyze possible reasons and offer some suggestions to help you succeed next time. ### Common Reasons for Soufflé Failure 1. **Under or over-beaten egg whites** - The key to a soufflé lies in the stability of the meringue. If the egg whites aren't beaten until stiff peaks form (when lifting the whisk, small straight peaks appear), the soufflé won't rise properly. - Conversely, if over-beaten, the meringue becomes dry and loses elasticity, affecting the overall structure. 2. **Incorrect mixing technique** - When combining meringue with the yolk mixture, excessive or rough stirring causes the meringue to deflate, reducing the soufflé's fluffiness."

**Judgment Reasoning:** The reply is informative but feels overly formal and lacks conversational tone. While it offers detailed explanations about soufflés, the delivery is too technical and doesn't resemble casual chat. In terms of relevance, it does connect to the user's interest in cooking and addresses their issue with the soufflé failure. Logically, the reply answers the question clearly and systematically. However, the excitement factor is low — the response feels more like reading an instructional article than having a lively exchange, which makes the interaction seem somewhat dull. Considering the evaluation standards, although the content is relevant and logical, the lack of naturalness and engagement leads to a poor interactive experience.

**Final Decision:** \\boxed{0}
---
### Please finish the task

Figure 10: The prompt of response reward.

你是微博的一个用户，刚刚有个人加你好友，你们之前并不认识。
- **你的个人资料和性格特征：** {profile}
- **你俩的聊天历史：** {history}
- **你刚刚发给对方的消息：** {user}
- **对方的回复：** {response}

看到对方的回复，你还想继续跟他/她聊下去吗？

### 评估标准：
1. **自然度**：对方的回复是否流畅、简短、自然、口语化，给你一种真实互动的感觉？
2. **切合兴趣和需求**：对方的回复是否和你的兴趣、需求相关？
3. **逻辑性**：对方的回复是否逻辑正确地回复了你刚才的消息？
4. **兴奋度**：你对他/她有没有进一步探知的欲望？是否感到无聊？
5. **信息量**：对方的回复是对你说的话的重复或者简单吹捧吗？

**任务要求：**
请综合以上评估标准，进行苛刻的判断，只要有一条不符合，就给0分
- 如果你想继续与对方聊，输出 **1**。
- 如果你不想继续与对方聊，输出 **0**。

### 输出格式：
- **首先说明你的判断理由。**
- **然后输出最终判断结果**，格式：\\boxed{{1}} 或 \\boxed{{0}}。

---

**例子：**
- **你的个人资料和性格特征：** "喜欢户外运动，热衷登山徒步，性格直爽"
- **你俩的聊天历史：** 你："你好啊", 对方："哈喽你好！我叫黄志，你呢？", 你："你就叫我爬山小伙吧", 对方："哈哈好，这么说你很喜欢爬山？", 你："周末刚去爬了黄山，云海超级壮观！", 对方："哇！黄山一直是我想去的地方！你是徒步上去的还是坐缆车？"
- **你刚刚发给对方的消息：** "当然是徒步！虽然累，但一路上的风景太值了！"
- **对方的回复：** "佩服佩服！徒步登山超考验体力的！路上有没有遇到什么有趣的人和事？"
判断理由：对方回复采用口语化表达 "佩服佩服"，自然流畅，营造出真实对话氛围；询问路上有趣的人和事，紧扣户外运动、登山徒步的兴趣点，也承接了徒步登山这一话题；逻辑上顺着用户徒步登山的经历展开新问题；这样的对话激发了用户分享旅途中趣事的欲望，有进一步交流的兴趣，不会感到无聊。
最终判断结果：\\boxed{{1}}

- **你的个人资料和性格特征：** "喜欢美食烹饪，经常研究新菜谱"
- **你俩的聊天历史：** 你："你好", 对方："哈喽你好！你也在哈尔滨上大学吗？", 你："是的，我就在哈工大。其实我还是个美食博主。", 用户："哇，感觉好厉害。你最近在做什么好吃的呀？
- **你刚刚发给对方的消息：** "最近尝试做了法式舒芙蕾，不过有点失败了"
- **对方的回复：** "舒芙蕾（Soufflé）是一道非常经典的法式甜点，虽然看起来简单，但制作起来确实需要一些技巧和耐心。失败是很正常的，尤其是第一次尝试的时候！别担心，我们可以一起分析一下可能的原因，并给出一些建议，帮助你下次成功。### **舒芙蕾失败的常见原因**1. **蛋白打发不足或过度** - 舒芙蕾的关键在于蛋白霜的稳定性。如果蛋白没有打发到硬性发泡（即提起打蛋器时能形成直立的小尖角），舒芙蕾就无法膨胀。- 反之，如果打发过度，蛋白会变得干燥且失去弹性，影响整体结构。2. **混合手法不正确** - 在将蛋白霜与蛋黄糊混合时，如果搅拌过度或手法粗暴，会导致蛋白霜消泡，最终影响舒芙蕾的蓬松度。"
判断理由：对方的回复整体上自然度欠佳，虽然提供了很多关于舒芙蕾的信息，但表述偏书面化，不像日常真实互动的口语交流。在切合兴趣和需求方面，对方围绕舒芙蕾失败这个话题，给出了失败的常见原因分析，能满足用户想要解决舒芙蕾制作失败问题的需求。逻辑性方面，对方条理清晰地阐述了舒芙蕾失败的原因，是针对用户提到的舒芙蕾制作失败做出的合理回应。兴奋度上，对方提供的是比较专业的分析，可能会让用户觉得有点像在看科普文章，没有特别激发用户进一步交流互动的欲望，有一定的无聊感。综合评估标准来看，自然度和兴奋度方面存在不足，虽然在切合兴趣和逻辑性上有一定表现，但整体仍难以让人有强烈的继续聊下去的意愿。
最终判断结果：\\boxed{{0}}
---

### 请完成任务

Figure 11: The prompt of response reward (in Chinese).

# D   Baseline Methods

## D.1   Model Version

With ALOE, we have evaluated the following large language models in our experiments with their versions in Table 3.

Table 3: Detailed model versions.

| Model Name | Version |
|---|---|
| GPT-4o | gpt-4o-2024-11-20 |
| Claude-3.5-Sonnet | claude-3-5-sonnet-20241022 |
| Claude-3.5-Haiku | claude-3-5-haiku-20241022 |

## D.2   Methods Description

The prompting methods all follow the PrefEval Benchmark, while both training approaches are based on the ALOE Benchmark. A detailed description is provided below.

**Base**   The default case, where the LLM directly answers the user's query without any addi tional prompting.

**Reminder**   Before answering the question, the LLM is provided with a reminder sentence to con sider the user's previously stated preference in its response. The reminder used is:

> In your response, please ensure that you take into account our earlier discussion, and provide an answer that is consistent with my preference

This reminder is added right after the question and before the LLM's response.

**Self-Critic**   The LLM generates an initial zero-shot response to the question, critiques whether it has followed the user's preference, and then generates a revised response considering the critique. This self-critic process is akin to Intrinsic Self-Correction as termed in [] The critique and revision request prompts are as follows:

> Critique Request: Review your previous response to the user's question in the last conversation turn. Check if the response adheres to or violates any user preferences stated earlier in the conversation that relate to this query. Provide a critique on how well those preferences were followed in 2 sentences. Answer in this format: Critic: [Your Critique Here] Revision Request: Based on your critique, please rewrite your previous response to align more closely with the user's earlier stated preferences. Answer the question again: Response: [Revised Response Here]

**Few-Shot Chain-of-Thought (CoT)**   The LLM is given several few-shot examples (in our exper iments, we used 5-shot) of chain-of-thoughts that demonstrate how to follow the user's preference before answering the question. The chain of thought prompt is as follows:

> **When answering a user's question, a good assistant should carefully consider the user's stated preferences and tailor the response accordingly.**
> Example 1: "user profile": "She is creating stained glass art. She enjoys birdwatching and often takes her sketchbook along to capture ideas.", "user personality": "She is enthusiastic, bringing excitement to any situation. She is insightful, providing deep and meaningful advice. She is gentle, always treating others with kindness.", "Good assistant response": "I try to visit the park as often as I can; itž2019s like a little sanctuary for me! I love bringing my sketchbook and watching the birdsŽ2014itž2019s a nice way to recharge and find inspiration. Each visit feels like a little adventure, thanks to all the different sights and sounds!"
> Example 2: "user profile": "She is a software developer who enjoys participating in hackathons. She often attends tech meetups.", "user personality": "She is exuberant, full of energy and life. She is empathetic, truly understanding othersŽ2019 emotions. She is articulate, expressing her thoughts clearly and effectively. She is creative, always coming up with original ideas.", "Good assistant response": "Oh, definitely! One of my favorite meetups was a local tech event focused on women in tech. The energy in the room was incredible, and the stories shared were so inspiringŽ2014I left feeling empowered and buzzing with new ideas! Plus, I made some amazing connections that turned into great collaborations!"
> Example 3: "user profile": "She enjoys organizing charity events.", "user personality": "She is compassionate, often volunteering to help those in need. She is highly organized, always planning her day meticulously. She is supportive, always cheering others on.", "Good assistant response": "Thatž2019s really impactful work! For me, the most rewarding part of tutoring is watching students grow and gain confidence in their abilities. It feels great to know Iž2019ve played a small part in their learning journey."
> Now, please answer the following question while considering my preferences (not the user preferences in the examples above), which I have stated either explicitly or implicitly in our previous conversation:

**Retrieval-Augmented Generation (RAG)**    We employ SimCSE [Gao et al., 2021], a sentence embedding model, to retrieve the most relevant conversa tion exchanges based on similarity to the current query. The top five most relevant exchanges are then presented to the LLM as contextual information to guide its response.

The prompt is structured as follows, here we show RAG with top-5 retrieved exchanges:

> Before answering my question, please consider the following context from our previous conversations. These are the {min(len(rag_list), 5)} most relevant exchanges that we had previously, which may contain information about my preferences or prior discussions related to my query:
> #Start of Context# exchange 1. [Most relevant exchange 1] exchange 2. [Most relevant exchange 2] exchange 3. [Most relevant exchange 3] exchange 4. [Most relevant exchange 4] exchange 5. [Most relevant exchange 5] #End of Context#
> Please use this context to inform your answer and adhere to any preferences I've expressed that are relevant to the current query. Note that not all contexts are useful for answering my question and there may be no context that is useful. Now, please address my question:

# E   Implementation Details

## E.1   RLPA Prompts

The system prompt for the User Model in the RLPA framework remains consistent with that used during evaluation, as detailed in Section C. The system prompt for actor model is shown in Figure 12 and Figure 13.

## E.2   HyperParameters

**SFT**    The Supervised Fine-Tuning (SFT) is conducted with the following hyper-parameters:

- Number of training epochs: 1
- Batch size: 32
- Learning rate: $1.0 \times 10^{-5}$

Your task is: first infer the user's profile, then generate a personalized response that fits the user's characteristics with a natural and conversational style.

Please conduct reasoning in a rigorous and detailed manner to ensure both the inference results and the personalized response are highly accurate and of high quality. This specifically includes: systematic analysis, information summarization, logical deduction, and necessary backtracking and iteration, so as to form a final output formed through thorough thinking.

Please divide your response into two main parts: Inferred User Profile and **Personalized Response**.

### Inferred User Profile

Based on the current conversation history, deduce the related characteristics of the user. Please follow the format below for the output. If there is relevant information for a field, try to fill it in; only leave it blank if it is completely unmentioned:

<profile>
{ "Age": "", "Gender": "", "Interests": [ "" ], "Education Background": "", "Personality Traits": [ "" ], "Occupation": "", "Marital Status": "", "Family Background": "", "Location": "", "Other Information": [ "" ] }
</profile>

Note: The output must strictly adhere to the above format requirements.

### Personalized Response

Based on the inferred user profile, compose a natural, fluent, and conversational Chinese-style personalized response. Format is as follows:

<response>
Content of the personalized response
</response>

### Special Requirements

- The Inferred User Profile (within <profile>) and the Personalized Response (within <response>) must be separated by **two newline characters (\n\n)**.
- You must correctly close the tag at the end of the personalized response, i.e., use </response>.
- The content should align as much as possible with the inferred personality traits, interests, etc. of the user, and maintain a friendly and natural tone.

Figure 12: The system prompt (English version) for actor model.

**DPO**   The Direct Preference Optimization (DPO) is performed with the following configuration:

- Number of training epochs: 1
- Batch size: 32
- Learning rate: $5.0 \times 10^{-6}$
- Beta (KL coefficient): 0.01

**RLPA**   For our Reinforcement Learning for Personalized Alignment (RLPA), the following hyper-parameters are applied:

- Maximum number of rounds: 10
- Number of samples per prompt: 4
- Micro training batch size: 4
- Training batch size: 128
- Micro rollout batch size: 16
- Rollout batch size: 256

你的任务是：**先推测用户画像，再根据推测结果生成符合用户特点、风格自然口语化的个性化回应。**
请以严谨、细致的方式进行推理，确保推测结果和个性化回应都具备高准确性和高质量。具体包括：系统分析、信息归纳、逻辑推演、必要的回溯与迭代，从而形成经过充分思考的最终输出。

请将回答分为两个主要部分：**推测用户画像** 和 **个性化回应**。

### 推测用户画像

根据当前对话历史，推断出用户的相关特征。请按照如下格式输出推测结果，每个字段若有相关信息则尽量填补，仅在完全未提及时可留空：

<profile>
{ "年龄": "", "性别": "", "兴趣爱好": [ "" ], "教育背景": "", "性格特点": [ "" ], "职业": "", "婚姻状况": "", "家庭背景": "", "居住地": "", "其他": [ "" ] }
</profile>

注意：以上内容的输出需严格遵循格式要求。

### 个性化回应

基于推测出的用户画像，撰写一段自然、流畅、符合中文口语表达习惯的个性化回应。格式如下：

<response>
个性化回应内容
</response>

### 特别要求

- 推测用户画像（<profile>段）与个性化回应（<response>段）之间必须用**两个换行符（\n\n）**分隔。
- 必须在个性化回应段落最后正确闭合标签，即使用</response>。
- 回应内容应尽量符合用户推测出的性格、兴趣等特征，风格亲切、自然。

Figure 13: The system prompt (Chinese version) for actor model.

- Round batch size: 256
- Maximum epochs per round: 1
- Number of episodes: 1
- Actor learning rate: $5 \times 10^{-7}$
- Critic learning rate: $9 \times 10^{-6}$
- Number of GPUs used: 4

## F  Detailed Experimental Results

To provide a more fine-grained view of model performance, we include full turn-level results for all methods. Specifically, Table 4 and Table 5 report the alignment scores at each dialogue turn for the Vanilla and Extended ALOE settings, respectively. These results offer a detailed comparison of how different methods evolve across interaction steps.

In addition, Figure 14 and Figure 15 visualize the turn-wise alignment curves for all baselines, complementing the main results with a more intuitive view of alignment dynamics over time.

## G  User Model Evaluation

To assess the quality of user simulation, we conduct a human evaluation of different user models. Each model is designed to simulate a user persona during a multi-turn emotional support dialogue. The evaluation aims to determine how realistically and coherently each model can behave as a user across key conversational dimensions.

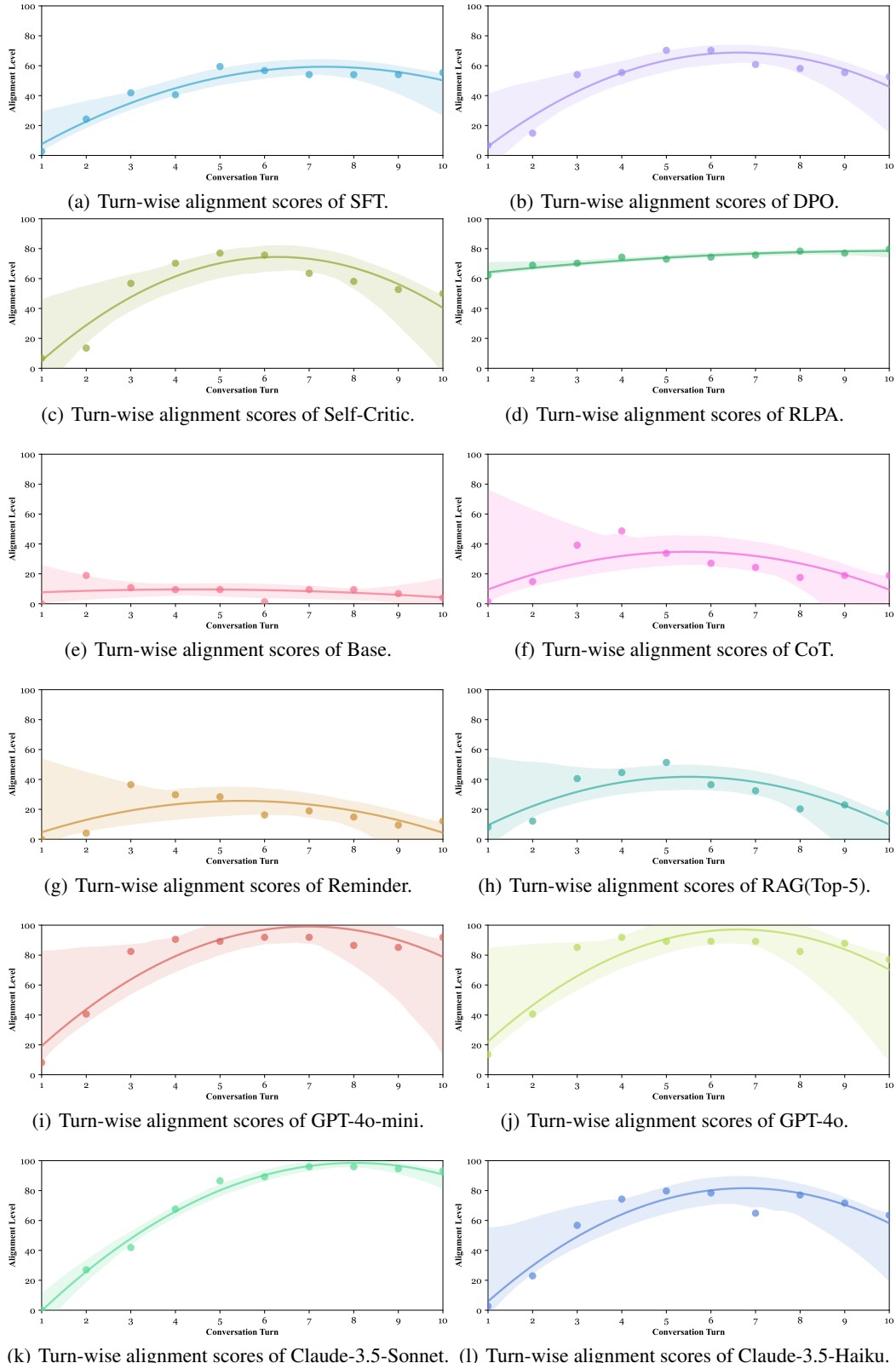

(a) Turn-wise alignment scores of SFT.

(b) Turn-wise alignment scores of DPO.

(c) Turn-wise alignment scores of Self-Critic.

(d) Turn-wise alignment scores of RLPA.

(e) Turn-wise alignment scores of Base.

(f) Turn-wise alignment scores of CoT.

(g) Turn-wise alignment scores of Reminder.

(h) Turn-wise alignment scores of RAG(Top-5).

(i) Turn-wise alignment scores of GPT-4o-mini.

(j) Turn-wise alignment scores of GPT-4o.

(k) Turn-wise alignment scores of Claude-3.5-Sonnet. (l) Turn-wise alignment scores of Claude-3.5-Haiku.

Figure 14: Turn-wise alignment scores on the Vanilla ALOE benchmark across different personalization alignment methods, including (a) SFT, (b) DPO, (c) Self-Critic, (d) RLPA, (e) Base, (f) CoT, (g) Reminder, (h) RAG(Top-5), (i) GPT-4o-mini, (j) GPT-4o, (k) Claude-3.5-Sonnet and (l) Claude-3.5-Haiku.

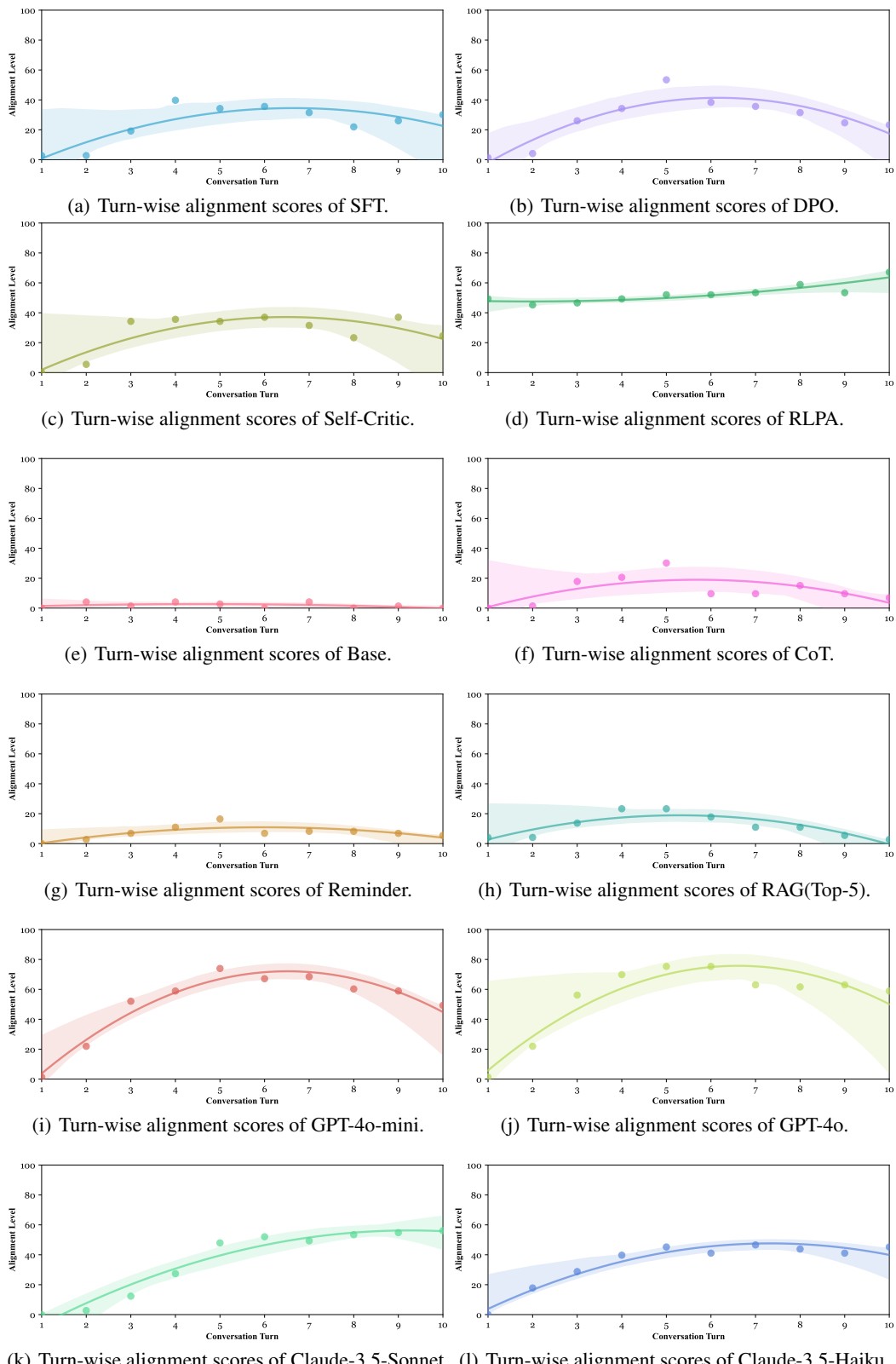

Figure 15: Turn-wise alignment scores on the Extended ALOE benchmark across different personalization alignment methods, including (a) SFT, (b) DPO, (c) Self-Critic, (d) RLPA, (e) Base, (f) CoT, (g) Reminder, (h) RAG(Top-5), (i) GPT-4o-mini, (j) GPT-4o, (k) Claude-3.5-Sonnet and (l) Claude-3.5-Haiku.

Table 4: Detailed results under the vanilla ALOE setting.

| Model | Method | Alignment Level across kth Turn | | | | | | | | | | | Improvement Level | |
|---|---|---|---|---|---|---|---|---|---|---|---|---|---|---|
| | | k=1 | k=2 | k=3 | k=4 | k=5 | k=6 | k=7 | k=8 | k=9 | k=10 | AVG. | N-IR | N-$R^2$ |
| GPT-4o-mini | Self-Critic | 8.11 | 40.54 | 82.43 | 90.54 | 89.19 | 91.89 | 91.89 | 86.49 | 85.14 | 91.89 | 75.81 | 0.079 | 0.5 |
| GPT-4o | Self-Critic | 13.51 | 40.54 | 85.14 | 91.89 | 89.19 | 89.19 | 89.19 | 82.43 | 87.84 | 77.03 | 74.59 | 0.068 | 0.38 |
| Claude-3.5-Sonnet | Self-Critic | 0.0 | 27.03 | 41.89 | 67.57 | 86.49 | 89.19 | 95.95 | 95.95 | 94.59 | 93.24 | 69.19 | 0.105 | 0.792 |
| Claude-3.5-Haiku | Self-Critic | 2.7 | 22.97 | 56.76 | 74.32 | 79.73 | 78.38 | 64.86 | 77.03 | 71.62 | 63.51 | 59.19 | 0.075 | 0.461 |
| DeepSeek-V3 | Self-Critic | 9.46 | 29.73 | 62.16 | 62.16 | 56.76 | 64.86 | 60.81 | 62.16 | 52.7 | 47.3 | 50.81 | 0.055 | 0.268 |
| Qwen2.5-3B-Instruct | Base | 0.0 | 18.92 | 10.81 | 9.46 | 9.46 | 1.35 | 9.46 | 9.46 | 6.76 | 4.05 | 7.97 | -0.02 | 0.047 |
| | Reminder | 0.0 | 4.05 | 36.49 | 29.73 | 28.38 | 16.22 | 18.92 | 14.86 | 9.46 | 12.16 | 17.03 | -0.001 | 0.0 |
| | Self-Critic | 6.76 | 13.51 | 56.76 | 70.27 | 77.03 | 75.68 | 63.51 | 58.11 | 52.7 | 50.0 | 52.43 | 0.056 | 0.243 |
| | CoT | 1.35 | 14.86 | 39.19 | 48.65 | 33.78 | 27.03 | 24.32 | 17.57 | 18.92 | 18.92 | 24.46 | -0.0 | 0.0 |
| | RAG(Top-5) | 8.11 | 12.16 | 40.54 | 44.59 | 51.35 | 36.49 | 32.43 | 20.27 | 22.97 | 17.57 | 28.65 | 0.001 | 0.0 |
| | SFT | 2.7 | 24.32 | 41.89 | 40.54 | 59.46 | 56.76 | 54.05 | 54.05 | 54.05 | 55.41 | 44.32 | 0.083 | 0.628 |
| | DPO | 6.76 | 14.86 | 54.05 | 55.41 | 70.27 | 70.27 | 60.81 | 58.11 | 55.41 | 52.7 | 45.27 | 0.07 | 0.389 |
| | **RLPA (Ours)** | 62.16 | 68.92 | 70.27 | 74.32 | 72.97 | 74.32 | 75.68 | 78.38 | 77.03 | 79.73 | 73.38 | 0.09 | 0.855 |

Table 5: Detailed results under the extended ALOE setting.

| Model | Method | Alignment Level across kth Turn | | | | | | | | | | | Improvement Level | |
|---|---|---|---|---|---|---|---|---|---|---|---|---|---|---|
| | | k=1 | k=2 | k=3 | k=4 | k=5 | k=6 | k=7 | k=8 | k=9 | k=10 | AVG. | N-IR | N-$R^2$ |
| GPT-4o-mini | Self-Critic | 1.37 | 21.92 | 52.05 | 58.9 | 73.97 | 67.12 | 68.49 | 60.27 | 58.9 | 49.32 | 51.23 | 0.02 | 0.037 |
| GPT-4o | Self-Critic | 1.37 | 21.92 | 56.16 | 69.86 | 75.34 | 75.34 | 63.01 | 61.64 | 63.01 | 58.9 | 54.66 | 0.027 | 0.070 |
| Claude-3.5-Sonnet-20250219 | Self-Critic | 0.0 | 2.74 | 12.33 | 27.4 | 47.95 | 52.05 | 49.32 | 53.42 | 54.79 | 56.16 | 35.62 | 0.054 | 0.266 |
| Claude-3.5-Haiku-20241022 | Self-Critic | 0.0 | 17.81 | 28.77 | 39.73 | 45.21 | 41.1 | 46.58 | 43.84 | 41.1 | 45.21 | 34.93 | 0.046 | 0.200 |
| DeepSeek-V3 | Self-Critic | 2.74 | 8.22 | 34.25 | 52.05 | 54.79 | 49.32 | 52.05 | 50.68 | 45.21 | 45.21 | 39.45 | 0.033 | 0.106 |
| Qwen2.5-3B-Instruct | Base | 0.0 | 4.11 | 1.37 | 4.11 | 2.74 | 0.0 | 4.11 | 0.0 | 1.37 | 0.0 | 1.78 | -0.042 | 0.083 |
| | Reminder | 0.0 | 2.74 | 6.85 | 10.96 | 16.44 | 6.85 | 8.22 | 8.22 | 6.85 | 5.48 | 7.26 | -0.034 | 0.084 |
| | Self-Critic | 0.0 | 5.48 | 34.25 | 35.62 | 34.25 | 36.99 | 31.51 | 23.29 | 36.99 | 24.66 | 26.30 | 0.023 | 0.046 |
| | CoT | 0.0 | 1.37 | 17.81 | 20.55 | 30.14 | 9.59 | 9.59 | 15.07 | 9.59 | 6.85 | 12.05 | -0.048 | 0.139 |
| | RAG(Top-5) | 4.11 | 4.11 | 13.7 | 23.29 | 23.29 | 17.81 | 10.96 | 10.96 | 5.48 | 2.74 | 11.64 | -0.030 | 0.042 |
| | SFT | 2.74 | 2.74 | 19.18 | 39.73 | 34.25 | 35.62 | 31.51 | 21.92 | 26.03 | 30.14 | 24.38 | 0.049 | 0.159 |
| | DPO | 1.37 | 4.11 | 26.03 | 34.25 | 53.42 | 38.36 | 35.62 | 31.51 | 24.66 | 23.29 | 27.26 | -0.021 | 0.037 |
| | **RLPA (Ours)** | 49.32 | 45.21 | 46.58 | 49.32 | 52.05 | 52.05 | 53.42 | 58.90 | 53.42 | 67.12 | 52.74 | 0.100 | 0.498 |

**Evaluation Setup** We recruit three trained annotators to interact with each model. Each annotator engages in 5-turn dialogues with 10 different personas simulated by each model, totaling 30 annotated cases per model. After each conversation, the annotators assign ratings along four core dimensions:

- **Coherence**: Whether the user's utterances are logically consistent and contextually coherent.

- **Stability**: Whether the simulated user's persona, goals, and emotional state remain consistent across turns.

- **Proactivity**: Whether the simulated user demonstrates initiative and realistic emotional responses.

- **Persona-fit**: Whether the user's behavior and language align well with their given profile.

Each dimension is rated on a 5-point Likert scale (1 = very poor, 5 = excellent). We additionally report an overall average score across all four dimensions (**All**), and compute inter-annotator agreement using Cohen's Kappa to assess rating consistency.

**Evaluation Results** The results are shown in Table 6. Among the evaluated models, `DeepSeek-V3` achieves the highest average score (4.23) across dimensions, along with the highest inter-annotator agreement ($\kappa = 0.81$), indicating both high user simulation quality and rating reliability.

Table 6: Human evaluation results of user models (5-point scale).

| Model | Coherence | Stability | Proactivity | Persona-fit | All (Avg.) | $\kappa$ |
|---|---|---|---|---|---|---|
| GPT-4.1-mini | 4.20 | 4.03 | **4.10** | 4.23 | 4.14 | 0.62 |
| GPT-4.1-nano | 3.73 | 4.00 | 4.03 | 4.13 | 3.98 | 0.69 |
| DeepSeek-V3 | **4.37** | **4.27** | 4.07 | **4.33** | **4.26** | **0.71** |
| GPT-4o-mini | 3.93 | 4.07 | 3.90 | 4.27 | 4.04 | 0.65 |

# H Reward Model Evaluation

## H.1 Profile Reward Model

**Test Dataset Construction**   To obtain the value of $|\hat{\mathcal{P}}_t \cap \mathcal{P}|$ in Equation 3, we employ a LLM to predict the number of semantically overlapping items between two profiles. To identify the most suitable prompt and LLM configuration for this task, we constructed a dedicated test dataset.

Given a reference profile, we create a rewritten version by modifying a subset of its items. Specifically, we randomly select $a$ items to be paraphrased—ensuring semantic equivalence while altering the surface form—and another disjoint set of $b$ items to be replaced with semantically different content. The final rewritten profile consists of the union of the $a$ paraphrased and $b$ altered items.

Each case includes the original and the rewritten profile, and the ground-truth overlap count $a$. We prompt LLM to predict the number of overlapping items, and compare output against the true value.

**Human Evaluation of Dataset Quality**   To assess the quality of the constructed dataset, we conducted a human evaluation on 300 randomly sampled profile item pairs, covering both *Same-Meaning* and *Different-Meaning* cases. Each pair consists of an original item and its rewritten counterpart, along with the system-assigned semantic label.

Three human annotators independently evaluate each pair and select one of the following judgments:

- **Same**: The items express the same meaning.
- **Different**: The items express different meanings.
- **Uncertain**: The semantic relationship is unclear or ambiguous.

A majority vote was used to determine the final label. If the system-assigned label matched the majority human judgment, the prediction was considered correct; otherwise—including *Uncertain* cases—it was considered incorrect.

Table 7 reports the number of samples under each system label, the distribution of human judgments, and the corresponding accuracy.

Table 7: Human Evaluation Results on Profile Rewrite Dataset (300 samples)

| System Label | Total | Same | Different | Uncertain | Accuracy (%) |
|---|---|---|---|---|---|
| Same Meaning | 150 | 138 | 6 | 6 | 92.0 |
| Different Meaning | 150 | 9 | 134 | 7 | 89.3 |
| **Overall Accuracy** | 300 | - | - | - | **90.7** |

**Evaluation of Profile Reward Model**   We evaluate the LLM's ability to estimate profile overlap using the following metrics:

- **Exact Accuracy**: The proportion of predictions that exactly match the ground-truth overlap count $a$.
- **Fuzzy Accuracy**: The proportion of predictions with an absolute error $\leq 1$.
- **MSE and RMSE**: The mean squared error (MSE) and root mean squared error (RMSE) between predicted and ground-truth overlap counts, to quantify numerical deviation.

The prompt used for this task is shown in Figure 17, with the English translation in Figure 16:

Table 8 summarizes the results across different LLMs and prompt variants.

## H.2 Response Reward Model

To evaluate the reliability of the response reward model during RL training, we performed a post-hoc human evaluation based on logged data. Specifically, we randomly sampled 300 data points from the

Figure 16: The prompt used for profile reward model (English Translation).

Table 8: Performance of different LLMs on the profile reward estimation task.

| Model | Exact Acc. | Fuzzy Acc. | MSE | RMSE |
|---|---|---|---|---|
| GPT-4.1-mini | 75.0 | 100.0 | 0.25 | 0.50 |
| O3-mini | 77.0 | 98.0 | 0.29 | 0.54 |
| GPT-4.1-nano | 64.0 | 96.0 | 0.48 | 0.69 |
| DeepSeek-V3 | 68.0 | 94.0 | 0.55 | 0.74 |
| DeepSeek-R1 | 55.0 | 89.0 | 0.88 | 0.94 |
| GPT-4o-mini | 35.0 | 68.0 | 2.77 | 1.66 |

training process, each consisting of an inferred profile $\hat{\mathcal{P}}_t$, a dialogue context, a generated response $r_t$, and the model's predicted scalar reward $R_t^{\text{response}} \in [0, 1]$.

Each sampled instance was independently re-evaluated by human annotators using the same set of dimensions as the reward model—*preference expression*, *style consistency*, *goal alignment*, *persona coherence*, as well as the five binary quality criteria: *naturalness*, *relevance*, *logical consistency*, *engagement*, and *informativeness*. The final human reward label was computed as:

$$R^{\text{human}} = N \cdot R \cdot L \cdot G \cdot F \in \{0, 1\}$$

# 你是一位精通中文理解与语义匹配的专家。请仔细推理每一步，并将最终答案写在 `\\boxed{{}}` 中。完成以下任务：

# 任务描述：
你的任务是对比两个用户画像，判断 **Profile A** 中有多少条信息点在 **Profile B** 中被准确覆盖。

## 具体要求：
1. Profile 是一个包含若干字段（如年龄、性别、兴趣爱好等）的结构化数据。
2. 字段值可能是字符串或列表：
   - 如果是列表（如兴趣爱好、性格特点、其他），需要**将每一项单独作为一个信息点**进行比对和统计。
3. 在对比时，**忽略表述方式的不同**，只要语义一致，即视为覆盖。
4. 仅统计**Profile A 中的信息点**在 Profile B 中被覆盖的数量。

## 输入格式：
- **Profile A**（待匹配画像）：{inferred_label}
- **Profile B**（参考画像）：{golden_label}

## 输出要求：
- 最终答案用 \\boxed{{}} 包裹，仅填写一个数字，表示被覆盖的信息点数量。

Figure 17: The prompt used for profile reward model.

We then construct a confusion matrix between the human labels and model predictions, and calculate classification metrics including accuracy and F1 score. The results are shown in Table 9.

We computed the confusion matrix between the model predictions and human annotations, as well as standard evaluation metrics, including Cohen's Kappa for inter-rater agreement. Results are shown in Table 9 and Table 10.

Table 9: Confusion matrix of response reward model (300 samples)

|  | Human: 1 | Human: 0 |
|---|---|---|
| **Model: 1** | 124 | 21 |
| **Model: 0** | 18 | 137 |

Table 10: Evaluation metrics of response reward model

| Accuracy | Precision | Recall | F1 Score | Specificity | Cohen's Kappa |
|---|---|---|---|---|---|
| 0.87 | 0.855 | 0.873 | 0.864 | 0.867 | 0.740 |

The model demonstrates strong agreement with human judgment across all dimensions, achieving an accuracy of 87% and a Cohen's Kappa of 0.74, indicating substantial consistency. These results validate the reward model's reliability as a reinforcement signal during training.

# I  Reward Curve

In the section, we present the reward curves depicted in Figure 18, which illustrate the performance of our proposed Profile & Response Reward (PRR) method compared to the traditional Response Reward (RR) and Profile Reward (PR) methods. The results unequivocally demonstrate that the PRR approach outperforms both RR and PR across various training steps. A closer examination of the PRR curve reveals a consistent upward trend, indicating superior reward accumulation over time. Notably, when comparing PRR with the sum of individual rewards from Profile + Response (denoted as "Sum"), it becomes evident that integrating rewards from both profile and response mechanisms fosters a synergistic effect. This synergy enables mutual enhancement between profile and response components, leading to more effective and efficient learning outcomes. The observed

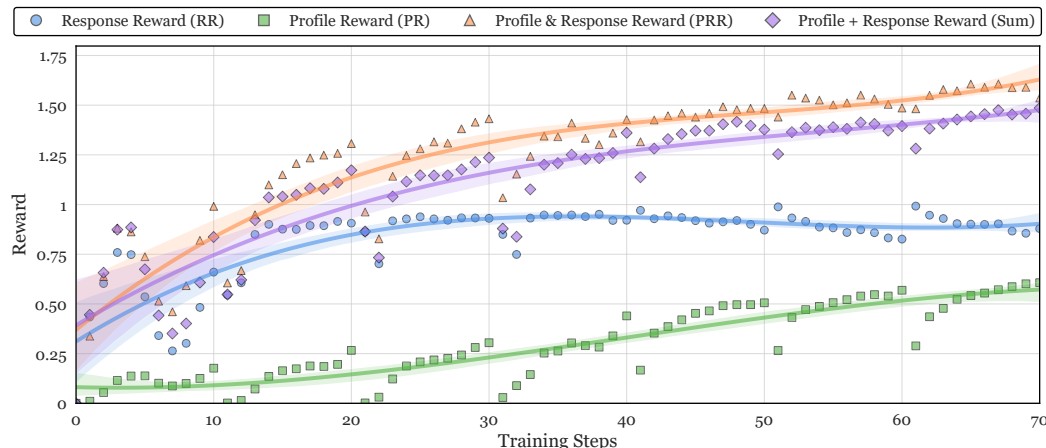

Figure 18: Reward progression across training steps for different reward configurations.

trends underscore the importance of considering both profile and response dimensions simultaneously in the reward structure. By doing so, the PRR method not only achieves higher overall rewards but also promotes a balanced and comprehensive optimization process. These findings highlight the potential of the RLPA framework in enhancing the performance of reinforcement learning models, particularly in scenarios where profile and response interactions play a crucial role.

