# OpenReview forum: "Teaching Language Models to Evolve with Users: Dynamic Profile Modeling for Personalized Alignment"
_NeurIPS.cc/2025/Conference — NeurIPS 2025 poster_

### Official Review · Reviewer_Nzdd · 2025-06-09

**Clarity:** 2
**Significance:** 3
**Originality:** 2
**Rating:** 4
**Confidence:** 4

**Summary:**

The paper addresses the problem of personalized alignment, where a language model (LM) is expected take into account user-specific information inferred from the conversation. They train the LM using Reinforcement Learning to extract then incorporate the preferences of a user simulated by another LM. The reward function is the combination of a profile reward function, which encourages the model to extract the latest user preferences  , and a response reward function, which incentivizes the model to align its responses to the extracted preferences. They experiment on  Qwen-2.5-3B-Instruct and outpeform prompting and fine-tuning baselines.

**Questions:**

1 - Do you only use exact match to compute the profile reward? How do you capture user preferences that are formulated in different terms from the labels?

2 - How exactly is the response reward computed? There is not enough details in the paper though this is a big part of the contribution of the paper.

3 - What is the performance on very long context? How does this method compare to others near the limit of the context window?

**Ethical Concerns:**

["NO or VERY MINOR ethics concerns only"]

**Final Justification:**

The authors have proposed answers to all the questions. This is especially true for the third question. The clarity of the paper was improved. I have increased the clarity score but I will maintain my overall score of 4.

**Limitations:**

yes

**Quality:**

3

**Strengths And Weaknesses:**

Strengths:
- Personalized alignment is an important problem and the paper proposes  a solution that is competitive with the closed models.
- The paper uses a controlled environment which makes evaluation more rigorous.
- The paper compares the proposed method with a wide range of baseline methods and models.

Weaknesses:
- The profile reward function may not fully capture the extraction of user preferences. The rewards is computed using a matching score. However, it is possible that the extracted preferences are correct but happen to be expressed in words different from the "ground-truth" user profile.
- Some crucial parts of the paper are not very clear. There is very little information about the response reward even in the appendix. The reward is based in four core dimensions and five other binary criteria. However, these concepts are not clearly defined (even in the appendix), and it is unclear how the reward model takes them into account to produce the scalar.
- The long-term interaction experiment might be limited. Models are evaluated on 70 turn dialogues, but it is unclear how much of the context window these dialogues actually occupy.

---

> ### Author Rebuttal · Authors · 2025-07-30
>
> Thank you very much for your thorough review and valuable feedback. We sincerely appreciate your recognition of our work’s strengths, including the importance of personalized alignment, the controlled evaluation setup, and the comprehensive comparison with baselines.
>
> We now address the concerns you raised.
>
> ---
>
> > **W1 & Q1: Limitations of the profile reward function**
>
> We sincerely appreciate this insightful observation. The concern is indeed valid, and we have taken measures to address it in our reward design. Specifically, we adopt a **slot-value structured matching** approach, rather than relying on surface-level word matching, to mitigate discrepancies caused by semantically equivalent but lexically different expressions. As described in §3.2 of our paper (Page 5, Lines 156–165).
>
> Moreover, during the construction of training and evaluation data, we utilize GPT-4o-mini to convert natural language profiles into structured JSON representations (Appendix D.2). This conversion enables **a certain degree of semantic normalization**, helping to reduce the sensitivity of the reward to superficial wording differences.
>
> That said, we acknowledge that this method is not foolproof and may still fail to capture all semantically equivalent expressions. To address this limitation, we plan to explore more robust semantic matching mechanisms—such as entailment-based or embedding-based similarity metrics—in future work.
>
> ---
>
> > **W2 & Q2: Lack of clarity on the response reward design**
>
> We apologize for the insufficient explanation and would like to provide further clarification below:
>
> As defined in §3.3 (Page 5, Lines 172–176), the response reward is computed in Equation 4.
>
> That is, a response must first satisfy all five **foundational quality criteria** (N, R, L, G, F) before any reward is assigned based on the four **personalization-related dimensions**. This gating mechanism ensures that only responses of sufficient quality are eligible for personalized alignment rewards, thereby increasing the discriminative power of the reward signal and preventing misleading gradients during training due to low-quality outputs.
>
> To further clarify, we provide the full bilingual (English and Chinese) **evaluation prompt** used by the reward model in **Appendix D.4**, which includes detailed definitions, judgment rules, and examples for each dimension and criterion. We will additionally include a reference to this appendix in the main text in the camera-ready version to improve clarity and coherence.
>
> ---
>
> > **W3 & Q3: Lack of clarity on the response reward design**
>
> This is an important and insightful concern — thank you for raising it. We have conducted an additional analysis to better quantify context usage in our 70-turn long dialogue setting. Our empirical statistics are as follows:
> - On average, **user utterances** are ~74 tokens per turn.
> - The **assistant’s response** is ~67 tokens, and
> - The **inferred profile**, which is explicitly generated each turn, is ~97 tokens.
>
> This results in an average of **~164 tokens per turn**, accumulating to over **16,000 tokens** across 70 turns — far exceeding the typical context window limit (8192 tokens). In practice, we **confirmed that context truncation does occur** in long interactions.
>
> However, we would like to emphasize a key design feature of RLPA that addresses this issue: the inferred profile is not only included in the generation output, but also **re-injected in a structured format** at the beginning of the next turn. This explicit profile tracking ensures that even when earlier context is truncated, the essential personalization information is retained and reintroduced, **preserving alignment quality over long dialogues**.
>
> We will include this empirical analysis and clarification in the revised version of the paper to more explicitly address concerns about the effectiveness of our method in long-term interactions.
>
> ---
>
> Once again, we are grateful for your thoughtful review, and we hope our responses address your concerns.

---

> > ### Comment · Reviewer_Nzdd · 2025-08-03
> >
> > Thank you for your answer. The authors have proposed answers to all the questions. This made the paper clearer especially the third question. However, the second point which is one the key contributions of the paper is still not very clear. I would say the level or rigor is not quite there yet. As a result, I will maintain my initial score.

---

> > > ### Author Response · Authors · 2025-08-04
> > >
> > > We sincerely thank the reviewer for the thoughtful comments and for pointing out the lack of clarity around the Response Reward design.
> > >
> > > To clarify, the Response Reward is computed using a scalar alignment score in [0,1], generated by a GPT-4o-mini reward model conditioned on the inferred user profile (Detailed prompt design is displayed in Appendix D.4). The reward model is guided by four core personalization dimensions—**preference expression, style consistency, goal alignment, and persona coherence**—which are reflected in its scoring prompt, but not evaluated as separate components.
> > >
> > > In addition to this scalar score, we implement **five binary criteria** to ensure basic response quality: **Naturalness (N), Relevance (R), Logical consistency (L), Engagement (G), and Informativeness (F)**. A response only receives a non-zero reward **if all five conditions are satisfied**, making the final reward. The motivation for these criteria emerged from failure cases we observed during early experiments: responses that superficially matched the user profile often suffered from robotic tone, vague content, logical inconsistency, or lack of engagement. The five binary filters were therefore introduced to **enforce a minimal standard of response quality** before rewarding profile alignment. This design also draws inspiration from prior work, especially the ALOE benchmark, which evaluates responses based on:
> > >
> > > - Whether the conversational style matches the user’s personality,
> > > - Whether the topic/content is relevant to the user’s profile,
> > > - Whether the response is human-like, engaging, and concise.
> > >
> > > Our binary criteria offer a **more fine-grained and enforceable decomposition** of these principles, enhancing both controllability and alignment robustness during RL training.
> > >
> > > We appreciate the reviewer’s insight in identifying this oversight, and we will revise the paper to better explain the role of these criteria, the prompt design of the reward model, and provide concrete examples to improve clarity.

---

> > > > ### Comment · Reviewer_Nzdd · 2025-08-09
> > > >
> > > > Thank you for taking the time to answer and for providing clarifications. The reward is an important part of the paper, so it should be explained as clearly as possible for the reader.

---

> > > > > ### Author Response · Authors · 2025-08-09
> > > > >
> > > > > We sincerely thank you for your thoughtful follow-up and for emphasizing the importance of a clear explanation of the reward design. We will incorporate the clarifications provided in our rebuttal directly into the revised manuscript to ensure readers can fully understand this aspect. We truly appreciate your constructive suggestions once again.

---

### Official Review · Reviewer_LN6q · 2025-07-02

**Clarity:** 4
**Significance:** 3
**Originality:** 2
**Rating:** 5
**Confidence:** 4

**Summary:**

This paper introduces RLPA, a reinforcement-learning framework to improve *personalized alignment* in language models. Specifically, models are trained to understand more accurate user profiles, containing information like user's location which are revealed in conversations, and to provide responses which are faithful to these profiles. Across the ALOE benchmark and compared to many baselines, the proposed method does well on Qwen-2.5-3B-Instruct, beating other baselines and remaining competitive with much larger models.

**Questions:**

* Would another option for profile reward be a free-text response, rather than a slot-value format? The generated profile could then be compared to the ground truth with a overlap metric or an embedding similarity metric
* I may be missing this, but it seems the rewards focus only on the generated profile and faithfulness to the profile. Did you try also including a reward component which measures how well the response matches the user-query itself? That seems to be an important component.

**Ethical Concerns:**

["NO or VERY MINOR ethics concerns only"]

**Final Justification:**

My original concerns were that although the proposed idea worked, it was limited in scope and tested only on one dataset and model. However, the authors wrote a extensive rebuttal which clarified major points in the paper and added another model, verifying the effectiveness of the method. Though the originality may not be outstanding, the impact is clear and I think this would be a good paper for the conference.

**Limitations:**

Yes.

**Paper Formatting Concerns:**

None.

**Quality:**

4

**Strengths And Weaknesses:**

Strengths
* The proposed method outperforms baselines on Qwen2.5-3B-Instruct, coming close to other much-larger baselines like GPT-4o.
* The proposed method is intuitive and well-justified. All parts of Section 3 are clear to follow and well-written.
* A variety of baselines are tested against
* The figures are consistently easy to read, informative, and pleasant to look at. This is not easy to consistently achieve. In particular, Figure 2 which introduces the method, is great at highlighting key information without overloading the reader. This continues in the Appendix, where figures are still clear and formatted well (such as the prompt examples)
* The appendix is extremely thorough with justifications for all experimental decisions
* Ablations are informative, showing the importance of multiple rewards and also comparing to other reasoning models

Weaknesses
* While the results are strong, they are ultimately done only on one dataset (ALOE) and with only one model. It is unclear if these results extend to other models/datasets. It would be helpful to try any of these: 1) test on other family of models 2) try on different size Qwen models for scaling purposes 3) Add another dataset, for evaluation (to show some kind of OOD generalization ability) or as a separate dataset entirely (do RL again)
* There are a couple of key terms which are not explicitly defined like "profile inference". While these definitions can be inferred, it would still be helpful to more explicitly mention them. In addition, the metrics have  in D.3 and I suggest justifying them at least briefly in the main text.

Comments
* x axis label on Figure 3 is a little too small. Also, the figure could be condensed to reduce white space - by combining them all into one graph.
* Can you make the colors a little more distinguishable from each other in Figure 4?
* Bolding may be unnecessary for Fig 5

---

> ### Author Rebuttal · Authors · 2025-07-30
>
> We sincerely thank the reviewer for the thoughtful and constructive feedback. We appreciate the recognition of our contributions. Below, we address the reviewer’s concerns point by point.
>
> ---
>
> > **W1: Limited evaluation on only one dataset and model**
>
> Thank you for raising this important point. We would like to clarify that our experiments are **not limited to a single dataset setting**. In addition to the original ALOE benchmark, we introduce an **Extended ALOE** setup that explicitly targets **out-of-distribution (OOD) generalization**.
>
> Specifically, Extended ALOE is constructed by sampling from PersonaChat and ensuring there is **no overlap in profile field types** between the training and test sets. This allows us to evaluate RLPA’s robustness to unseen profile structures and out-of-domain dialogue formats.
>
> To further address your concern, we **add the main experiments using the gemma-3-4b model**.
>
> - Vanilla ALOE
>
> |Model|Method|k=1|k=2|k=3|k=4|k=5|k=6|k=7|k=8|k=9|k=10|AVG.|N-1R|R²|
> |-|-|-|-|-|-|-|-|-|-|-|-|-|-|-|
> |Base|0|6.85|13.7|24.32|22.97|36.49|27.4|23.29|35.14|26.03|21.62|0.084|0.630|
> |Reminder|0|6.76|17.57|47.3|55.41|48.65|45.95|50.00|36.49|43.24|35.14|0.082|0.486|
> |Self-Critic|4.05|21.62|66.22|71.62|67.57|66.22|60.81|64.86|50.00|48.65|52.16|0.050|0.21|
> |CoT|4.05|10.81|41.89|43.24|40.54|29.73|27.03|31.08|16.22|25.68|27.03|0.018|0.027|
> |RAG|10.81|18.92|52.7|67.57|56.76|43.24|51.35|50.00|43.24|40.54|43.51|0.039|0.151|
> |SFT|4.05|14.86|56.76|71.62|66.22|52.70|62.16|48.65|44.59|37.84|45.95|0.039|0.130|
> |DPO|6.76|21.62|51.35|50.00|60.27|55.41|58.11|71.62|44.59|49.32|46.91|0.062|0.408|
> |RLPA|64.86|72.97|64.86|71.62|75.68|81.08|81.08|86.49|89.19|90.54|77.84|0.115|0.908|
>
> - Extended ALOE
>
> |Model|Method|k=1|k=2|k=3|k=4|k=5|k=6|k=7|k=8|k=9|k=10|AVG.|N-1R|R²|
> |-|-|-|-|-|-|-|-|-|-|-|-|-|-|-|
> |Base|0|4.05|6.76|13.7|15.07|15.07|13.7|8.22|10.96|10.96|9.85|0.062|0.314|
> |Reminder|0|5.48|13.7|27.4|26.03|23.29|24.66|26.03|27.4|23.29|19.73|0.092|0.593|
> |Self-Critic|2.74|8.22|53.42|54.79|46.58|36.99|39.73|35.62|34.25|20.55|33.29|0.023|0.042|
> |CoT|0|1.37|10.96|19.18|35.62|27.4|35.62|28.77|27.4|24.66|21.1|0.091|0.572|
> |RAG|0|17.81|32.88|50.68|41.1|27.4|26.03|39.73|27.4|24.66|28.77|0.028|0.096|
> |GFT|0|19.18|32.88|53.42|41.1|32.88|30.14|37.84|31.08|24.66|30.32|0.029|0.107|
> |DPO|1.37|17.81|36.99|61.64|62.16|53.42|46.58|54.79|49.32|43.84|42.79|0.064|0.361|
>
> RLPA consistently outperforms all baselines under both settings. These results strengthen our claim that RLPA is model-agnostic and generalizes well across profile structures and data distributions. We will include these results in the updated version and further discuss future directions for broader model family and dataset evaluation.
>
> ---
>
> > **W2: Missing definition of profile inference and justification of evaluation metrics**
>
> Thank you for pointing this out. We acknowledge that the term “profile inference” was introduced without an explicit definition in the main text, which may hinder clarity for some readers. To address this, we will revise the beginning of Section §3.2 to include the following clarification:
>
> *Profile inference refers to the model’s ability to estimate the user’s latent profileat each turn of the dialogue, based on prior conversational history. This predicted profile is then compared to the ground-truth profile in a slot-wise manner.*
>
> Regarding the evaluation metrics, we employ three normalized measures—**Align. Score**, **N-IR (Normalized Improvement Rate)**, and **N-R² (Normalized R-squared)**—to assess response alignment with user preferences. Their motivations and formulas are thoroughly detailed in Appendix D.3:
> - Align. Score is the primary metric and is computed using GPT-4o-mini to assess alignment between the model’s response and the ground-truth user preference.
> - N-IR accounts for variance in baseline performance across users and contexts.
> - N-R² measures the goodness of fit for alignment behavior across turns, reflecting consistency with the inferred profile.
>
> We agree that briefly explaining the rationale behind these metrics in the main text would improve accessibility. We will update §4.1 (“Metrics”) to include concise justifications for all three metrics.
>
> We appreciate your attentive review and helpful suggestions, which will help us strengthen the clarity and rigor of the final version.
>
> ---
>
> > **Q1: Alternative reward formats (e.g., free-text profile generation)**
>
> We sincerely thank the reviewer for raising this insightful and forward-looking question. While free-text profile generation offers greater flexibility, we opted for the **slot-value format** in this work based on both theoretical and empirical considerations:
>
> **1. Slot-value format is more structured, user-centric, and interpretable**
>
> Free-text profiles, though expressive, often result in verbose, redundant descriptions that accumulate surface-level preferences from recent turns, rather than capturing core, long-term user attributes. Consider the following comparison:
>
> - Free-text profile:
>
> *Prefers to grab food with friends after surfing; has a favorite beachside taco place with exceptional fish tacos and mango salsa; enjoys pairing food with drinks, especially cold coconut water…*
>
> - Slot-value profile
>
> {
>   "Age": "30",
>   "Interests": ["Surfing", "Collecting sneakers"],
>   "Personality Traits": ["Adventurous", "Ambitious", "Sociable"],
>   "Occupation": "Personal trainer",
>   "Others": ["Has a pet snake named Sly"]
> }
>
> The slot-based format allows for concise abstraction, supports long-term user modeling, and enables direct reward computation, which is crucial for stable RL training.
>
> **2. Slot-based profiles yield better performance in profile-matching tasks**
>
> We conducted additional experiments comparing slot and free-text profiles across multiple LLMs (e.g., GPT-4.1-mini, O3-mini, DeepSeek). The results (below) show **slot-value formats outperform free-text** across all metrics, including Exact Accuracy, Fuzzy Accuracy, MSE, and RMSE:
>
> - Slot
>
> |Model|Exact Acc.|Fuzzy Acc.|MSE|RMSE|
> |-|-|-|-|-|
> |GPT-4.1-mini|75.0|100.0|0.25|0.50|
> |O3-mini|77.0|98.0|0.29|0.54|
> |GPT-4.1-nano|64.0|96.0|0.48|0.69|
> |DeepSeek-V3|68.0|94.0|0.55|0.74|
> |DeepSeek-R1|55.0|89.0|0.88|0.94|
> |GPT-4o-mini|35.0|68.0|2.77|1.66|
>
> - Free-text
>
> |Model|Exact Acc.|Fuzzy Acc.|MSE|RMSE|
> |-|-|-|-|-|
> |deepseek-R1|65.0|92.0|0.77|0.88|
> |deepseek-V3|56.0|86.0|1.23|1.11|
> |gpt-4o-mini|56.0|85.0|1.72|1.31|
> |o3-mini|54.0|80.0|2.27|1.51|
> |gpt-4.1-mini|39.0|64.0|6.60|2.57|
> |gpt-4.1-nano|32.0|60.0|8.90|2.98|
>
> Free-text matching is further challenged by expression variability, redundancy, and noisier semantic signals.
>
> **3. Slot-value profiles improve alignment and training stability in RL**
>
> We trained RLPA using both formats under the same settings. As shown below, the slot-based version significantly outperforms the free-text version in alignment score (AVG), final step accuracy (k=10), and training stability (N-R²):
>
> |Model|Method|k=1|k=2|k=3|k=4|k=5|k=6|k=7|k=8|k=9|k=10|AVG.|N-1R|R²|
> |-|-|-|-|-|-|-|-|-|-|-|-|-|-|-|
> |Qwen2.5-3B-Instruct|Base|0.0|18.92|10.81|9.46|9.46|1.35|9.46|9.46|6.76|4.05|7.97|-0.020|0.047|
> |Qwen2.5-3B-Instruct|RLPA(free text)|46.58|51.35|54.05|48.65|58.11|59.46|64.86|62.16|60.81|56.76|56.28|0.082|0.577|
> |Qwen2.5-3B-Instruct|RLPA(slot)|62.16|68.92|70.27|74.32|72.97|74.32|75.68|78.38|77.03|79.73|73.38|0.090|0.855|
>
> This highlights the benefit of slot-value representation in maintaining long-term consistency and efficient optimization during reinforcement learning.
>
> **4. Embedding-based or ROUGE-based reward alternatives are less reliable**
>
> We also experimented with free-text alignment metrics using ROUGE-L and text embedding similarity. ROUGE-L proved fragile to paraphrasing:
>
> “Occupation”: “Personal trainer” vs. “Fitness coach” → ROUGE-L ≈ 0
>
> Embedding methods (e.g., OpenAI’s text-embedding-3-large) showed acceptable but **inferior performance** and required manual threshold tuning:
>
> |Model|Exact Acc.|Fuzzy Acc.|MSE|RMSE|
> |-|-|-|-|-|
> |text-embedding-3-small|54.0|86.0|0.96|0.98|
> |text-embedding-3-large|61.0|90.0|0.79|0.89|
>
> Moreover, the added computation and hyperparameter overheads make them less suitable for multi-turn, online RL optimization.
>
> **Conclusion**: While free-text profile generation is an interesting future direction, our experiments demonstrate that **slot-value formats are more interpretable, scalable, and empirically effective** for structured profile modeling and reinforcement learning. We will clarify this design choice and include these new findings in the revised version.
>
>
> ---
>
> > **Q2: Reward component for query relevance**
>
> In fact, this was explicitly considered in the design of our **Response Reward** function (§3.3).
>
> Specifically, we employed GPT-4o-mini to score generated responses across five binary dimensions:
> - Naturalness (N)
> - Relevance (R)
> - Logical consistency (L)
> - Engagement (G)
> - Informativeness (F)
>
> The final reward is computed as the product of all five criteria:
> (as described on **Line 175 of §3.3**). This means that a response must satisfy **all five aspects simultaneously** to receive a positive reward.
>
> Importantly, the **Relevance (R)** dimension directly evaluates whether the model’s response **appropriately addresses the user’s current input**, rather than merely repeating or restating user profile information.
>
> Therefore, our reward mechanism is already designed to capture both **faithfulness to inferred user profiles** and **responsiveness to the user query**, ensuring a holistic alignment objective.
>
> ---
>
> > **Comments on Figures**
>
> We sincerely thank you for the detailed and thoughtful suggestions regarding the figures. We appreciate your comments on improving the clarity and layout, and we will incorporate all your suggestions in the final version.
>
> ---
>
> Once again, we appreciate your close reading and valuable suggestion, and we hope this clarification resolves your concern.

---

> ### Comment · Area_Chair_C1JP · 2025-08-06
>
> Please read author rebuttal, thanks.

---

> ### Author Response · Authors · 2025-08-08
> **Kind Reminder to Reviewer LN6q**
>
> Dear Reviewer LN6q,
>
> Thank you again for your thoughtful feedback and valuable comments on our paper. We just wanted to kindly follow up and check if our rebuttal and the additional experiments have sufficiently addressed your concerns.
>
> If there are any remaining questions or clarifications we can provide, we’d be more than happy to assist.
>
> We truly appreciate your time and consideration.

---

### Official Review · Reviewer_ZvqW · 2025-07-03

**Clarity:** 3
**Significance:** 4
**Originality:** 3
**Rating:** 5
**Confidence:** 4

**Summary:**

This paper introduces the Reinforcement Learning for Personalized Alignment (RLPA) framework, a novel approach for teaching large language models (LLMs) to dynamically infer and adapt to user profiles during dialogue. The core contribution is a dual-reward RL system where a "Profile Reward" guides accurate user modeling and a "Response Reward" incentivizes the generation of personalized responses consistent with the inferred profile. The authors instantiate this by fine-tuning a Qwen-2.5-3B model, demonstrating state-of-the-art performance on personalized dialogue benchmarks. Notably, their model shows superior dynamic adaptation, long-term coherence, and reasoning efficiency compared to both prompt-based/offline-optimization baselines and larger commercial models.

**Questions:**

On the Simulation-to-Reality Transfer: The reliance on a simulated user is a key methodological choice. Could you elaborate on whether you did some explorations into this direction?

Handling Implicit and Ambiguous Signals: The experiments focus on relatively explicit user statements. How robust is the RLPA framework in inferring preferences from more implicit, ambiguous, or even sarcastic user utterances? A positive response would involve discussing whether the reward models are capable of capturing such nuance or if this represents a key area for future work.

**Ethical Concerns:**

["NO or VERY MINOR ethics concerns only"]

**Final Justification:**

Still seems like a great paper to me, I maintain my score.

**Limitations:**

Yes, the authors have adequately addressed the primary limitations and potential negative societal impacts of their work in Appendix A and B. They correctly identify the single-session nature of the framework and the inherent ethical risks of dynamic user profiling, such as privacy invasion and potential for manipulation. The discussion is transparent and constructive.
A suggestion for future work could be to explicitly propose mechanisms for user-controllable personalization. For example, allowing a user to directly view, edit, or delete parts of their inferred profile and define domains of their lives that should not be profiled.

**Paper Formatting Concerns:**

None.

**Quality:**

3

**Strengths And Weaknesses:**

Strengths:

1.  Significance and Originality: The paper makes a valuable contribution by tackling the important challenge of dynamic personalized alignment. The proposed Reinforcement Learning for Personalized Alignment (RLPA) framework offers a promising direction for moving beyond static alignment methods. Framing personalization as an interactive learning process is a thoughtful approach that appears more aligned with the dynamic nature of user dialogue than existing paradigms.

2.  Methodological Rigor and Quality: The research is well-designed and executed with care. The authors show good methodological practice by:
    *   Systematically designing and exploring their dual-reward models, including a clear rationale for their choice of LLMs in different roles.
    *   Utilizing a nuanced set of evaluation metrics (AVG, N-IR, N-R²) which provides a more complete picture of model behavior than a single performance score, effectively capturing the dynamics and stability of the learning process.
    *   Including helpful ablation studies that successfully demonstrate the individual contributions of the Profile and Response rewards to the overall framework.

3.  Robust and Comprehensive Evaluation: The experimental design is a clear strength of the paper. The decision to evaluate on both "in-format" and "cross-format" data provides a solid test of the model's ability to generalize. The inclusion of additional tests, such as the preference conflict and long-term interaction scenarios, adds considerable depth to the evaluation and provides compelling evidence for the model's adaptive capabilities in contexts relevant to real-world use.

Weaknesses:

1.  Simulation-to-Reality Gap: A primary limitation of the work is its reliance on a simulated user environment for both training and evaluation. While this is a practical necessity for the RL approach presented, it leaves open the question of how well these results will transfer to interactions with real human users. The paper commendably acknowledges this limitation.

2.  Scope of Personalization: The current framework focuses on a single, continuous dialogue session. As the authors note, important aspects of real-world personalization, such as multi-session memory and the ability to generalize user preferences across different domains, are not yet addressed by this work and remain interesting avenues for future exploration.

---

> ### Author Rebuttal · Authors · 2025-07-30
>
> We sincerely thank the reviewer for their thoughtful and constructive feedback. We are glad that the reviewer found our work to be significant, methodologically rigorous, and comprehensively evaluated. We appreciate the recognition of our core contributions, including the novel formulation of dynamic personalized alignment as a reinforcement learning problem, the design of our dual-reward RLPA framework, and the robustness of our empirical validation across multiple dimensions.
>
> Below, we address the specific concerns and questions raised:
>
> ---
>
> > **W1: Simulation-to-Reality Gap**
>
> We appreciate the reviewer’s insightful comment regarding the limitation of relying on a simulated user environment for both training and evaluation. We fully agree that, while simulation offers a necessary and controllable setup for developing our reinforcement learning framework, it cannot fully capture the complexity and diversity of real human behavior. We have acknowledged this limitation in Appendix A of the paper and emphasized the importance of bridging the gap between simulation and real-world interactions in future work.
>
> To address this challenge, we plan to incorporate real user interactions in subsequent experiments to better assess the model’s performance in practical dialogue settings. We believe that combining both simulated and real user feedback will enable us to further refine the model and move closer to real-world deployment requirements. Once again, we thank the reviewer for highlighting this critical point and will continue to investigate effective simulation-to-reality transfer strategies in our future research.
>
> ---
>
> > **W2: Scope of Personalization**
>
> Your observation aligns closely with our own thinking. From the outset of developing this work, we recognized the importance of long-term and cross-session personalization. RLPA, by design, focuses on dynamic adaptation within a single continuous dialogue session. More realistic deployment scenarios—such as multi-session interactions and persistent user profiles—indeed require additional engineering support, such as the integration of user-centered databases or memory modules.
>
> That said, we believe RLPA is well-suited to work in tandem with such external systems. Specifically, RLPA can be viewed as a short-term memory mechanism that captures immediate dialogue dynamics, while external user profile repositories can serve as long-term memory—analogous to the relationship between cache and disk in computer systems. This complementary setup has the potential to enable truly adaptive and persistent personalization.
>
> We thank the reviewer again for highlighting this important dimension. Your feedback is highly valuable and will help shape the next stage of our research.
>
> ---
>
> > **Q1: On the Simulation-to-Reality Transfer**
>
> We fully agree that bridging the gap between simulation and real-world deployment is a critical direction for future research.
>
> As described in Section 3 of the paper (Lines 122–151), our current approach uses constructed simulated user models for both training and evaluation. This design choice was made for several practical reasons:
> - **Controllability and reproducibility**: Simulated users allow us to precisely configure user attributes and control the gradual revelation of user preferences throughout the dialogue. This setup is particularly important for building stable and informative reward signals (both profile and response rewards) during reinforcement learning.
> - **Data efficiency**: Unlike human-in-the-loop settings, simulated interactions incur no annotation cost and enable the generation of large-scale, high-quality, and diverse dialogue traces. This supports more effective cold-start learning and long-range adaptation.
>
> That said, we are fully aware of the **simulation-to-reality gap**, and we have explicitly acknowledged this in the limitations section (Lines 486–490):
>
> In addressing the reviewer’s concern, we would like to highlight that **user simulation itself is a promising and underexplored research area**. Conceptually, simulated users can be viewed as role-playing agents whose behavior is governed by implicit identity traits, preferences, and personalities that unfold naturally through dialogue. We believe recent advances in **role-playing LLMs** can be effectively adapted to improve user simulation. Possible directions include:
> - Designing user behavior policies through prompt engineering (as we do with step-wise preference revelation);
> - Building structured user-profile-based dialogue datasets to train user models via supervised fine-tuning (SFT);
> - Applying reinforcement learning to user models themselves, optimizing for behavioral consistency and human-likeness.
>
> We believe that modeling user simulators as standalone research entities will not only improve the fidelity and generalizability of RL training environments, but also facilitate progress in user modeling, intent understanding, and personalized evaluation. Moving forward, we also plan to evaluate and extend RLPA in more open, mixed real-user settings.
>
> Thank you again for your thoughtful and constructive feedback!
>
> ---
>
> > **Q2: Handling Implicit and Ambiguous Signals**
>
> Thank you for highlighting this important challenge. We fully agree that the ability to handle implicit, ambiguous, or even sarcastic user expressions is crucial for real-world personalized dialogue systems.
>
> 1. Regarding the concern that our experiments focus mainly on explicit user preferences: while our main evaluation uses structured slot-value profiles (e.g., in the Vanilla and Extended ALOE settings), we intentionally designed our simulated users to **reveal preferences gradually and indirectly** (Section 3.1, Page 5). This setup encourages the model to infer user intent from weak or partial signals.
>
> Importantly, our reward models do **not rely on explicit user statements**.
> - **Profile Reward** is based on the accuracy of inferred slot values, rewarding correct inferences even from indirect or stylistically varied cues.
> - **Response Reward** assesses alignment with the inferred profile, emphasizing high-level criteria such as stylistic consistency, goal alignment, and emotional resonance (Section 3.2–3.3, Page 6).
>
> In Section 5.3, our long-term reasoning experiments show that RLPA can accumulate weak signals over multiple turns and improve profile inference progressively—demonstrating robustness to implicit cues.
>
> 2. That said, we acknowledge that **sarcasm and pragmatic deviations** (e.g., irony, humor) are not systematically covered in the current simulated environment. This is a limitation we note in the “Limitations and Future Work” section (Page 13). In future work, we plan to introduce richer pragmatic diversity—including sarcastic and ambiguous utterances—via human inputs or enhanced user simulators, to evaluate and improve RLPA’s capacity in handling such nuances.
>
> We sincerely thank the reviewer for drawing attention to this valuable direction.
>
> ---
>
> Once again, we thank the reviewer for their valuable comments, which have helped us better articulate the contributions and limitations of our work.

---

> ### Comment · Area_Chair_C1JP · 2025-08-06
>
> Please read author rebuttal, thanks.

---

### Official Review · Reviewer_Mt8V · 2025-07-04

**Clarity:** 3
**Significance:** 3
**Originality:** 2
**Rating:** 3
**Confidence:** 4

**Summary:**

This paper addresses two central challenges in personalized response generation: dynamically tracking users’ incrementally revealed profiles and generating high-quality responses accordingly. To tackle these issues, the authors introduce the Reinforcement Learning for Personalized Alignment (RLPA) framework with the **Profile Reward** and the **Response Reward**, and conduct thorough experiments to validate their approach.

**Questions:**

See Above.

**Ethical Concerns:**

["NO or VERY MINOR ethics concerns only"]

**Final Justification:**

As I mentioned in the Rebuttal, the quality of the paper deserves more than a 3 score, but the similar between the ALOE and this paper raised my concern. Of course, it is not about the technique, and therefore I give the 3 score. If the paper is finally accepted, it must make the essential revisions to the same content.

**Limitations:**

As the author also finds that the quality of the user simulator does not satisfy the experiments, I suggest that the author conduct more research on how to maintain a higher quality of the user simulator in multi-turn conversations.

**Paper Formatting Concerns:**

N/A.

**Quality:**

2

**Strengths And Weaknesses:**

**Strengths**

1. The paper presents a well-articulated motivation and thorough background, along with a precise analysis of the limitations in prior work.
2. The experimental evaluation is both comprehensive and rigorous, covering realistic scenarios. The appendix further strengthens the contribution by offering detailed accounts of preliminary studies.
3. The strong experimental results underscore the effectiveness of the two proposed reward functions in enhancing the quality of personalized responses.

**Weaknesses**

1. While the overall framework is clearly introduced, the internal mechanics of the method lack sufficient detail. In particular, the paper does not adequately explain how the user simulator functions or why it is effective. The process by which the assistant model aligns with the user profile through conversation is also underexplored. Although such mechanisms may have been described in ALOE, this paper omits many critical aspects, leaving readers unclear on how the assistant infers user attributes across turns—whether through proactive questioning or topic-specific interactions. It is widely understood that assistant models typically react to user input or assist with tasks, so the ability to actively enrich the user profile needs further clarification.
2. The connection between the structured user profiles used by the simulator and assistant, and their actual use in dialogue, is not clearly explained. For example, it is unclear whether the structured profiles come from a fixed, finite set that includes the attributes listed in the appendix. If so, this should be explicitly stated; if not, it is important to explain how the profiles’ quality is ensured. This is particularly relevant for practical applications, and the formulation resembles the slot-filling concept in task-oriented dialogue, as the authors themselves suggest.
3. The explanation of the evaluation protocol—including metric computation (e.g., AVG), baseline systems, and datasets—is insufficient. Although some of this information may have been provided in ALOE, large portions of the current manuscript, including both the main text and appendix, closely replicate content from the previous work. Even for background sections such as the introduction of datasets or metrics, direct reuse of previous text without substantial modification or citation is not acceptable and undermines the novelty of this submission. This is a very critical issue.
4. There are several issues in the experimental setup and its alignment with the paper’s main claims. (1) Although the user simulator is central to the proposed framework, the experiments do not convincingly demonstrate that it progressively reveals user attributes over the course of the conversation. Human evaluation only spans five turns, while the main results involve ten turns, leaving the effectiveness of later dialogue stages ambiguous. (2) For the Profile Reward, Equation (3) appears to compute a cumulative F1 score up to the current turn, which is non-incremental. If the simulator fails to express certain attributes, the recall remains low throughout, leading to consistently low scores, even when the LLM produces accurate outputs. (3) Although many baseline methods are included, the paper omits closely related work, such as AutoPal [1], which also dynamically infers user profiles for personalized response generation.

In addition, minor issues need to be addressed, such as Line 109 contains a typo—$\sum_{t=1}^{T}$ is incorrectly formatted.

[1] Cheng Y, Liu W, Xu K, et al. AutoPal: Autonomous Adaptation to Users for Personal AI Companionship. arXiv preprint arXiv:2406.13960, 2024.

---

> ### Author Rebuttal · Authors · 2025-07-31
>
> We sincerely thank the reviewer for the thoughtful and constructive feedback. We are pleased that the reviewer found our motivation well-articulated, appreciated the comprehensive experimental evaluation, and acknowledged the effectiveness of our proposed reward functions in enhancing personalized response quality.
>
> ---
>
> > **W1: Insufficient detail on user simulator and profile inference process**
>
> We thank the reviewer for raising this important point.
>
> Regarding the **user model**, we would like to clarify that the design and behavior of the simulated user are thoroughly described in Section 3.1 “Simulated User Design” (**Lines 129–150**). We also validate these properties through **human evaluations (Lines 147–150)**, which led us to select **GPT-4o-mini** as the final user model. Additional results and evaluations of the user model are provided in the **Appendix H**, and also further elaborated in our response to Weakness 4.
>
> Regarding the **assistant model**, our method is **RL-driven**—we do not manually design heuristics for inferring user profiles. Instead, the model is encouraged to **autonomously discover effective strategies** for personalization through simple reward signals. That said, we appreciate the reviewer’s concern and have conducted further analysis of the trained dialogue transcripts.
>
> Interestingly, we observed the following learning dynamics:
> - In early training stages, assistant models with few RL updates tend to **directly ask for user attributes**, which often results in unnatural or impolite behavior from the user model’s perspective.
> - After further training, the assistant adopts a more **indirect and effective strategy**, such as **self-disclosure**—proactively introducing related topics that implicitly reflect user preferences and observing the user’s response. This not only improves personalization performance (as measured by profile accuracy) but also maintains high response quality and conversational naturalness.
>
> We have included qualitative examples and a brief discussion of this behavior in **Appendix A.7** to further clarify how the assistant infers profiles in a realistic, adaptive manner.
>
> ---
>
> > **W2: Connection between structured profiles and dialogue use unclear**
>
> We thank the reviewer for highlighting this crucial point. Below, we clarify two key aspects of our user profile construction and its integration into the dialogue system:
>
> **1. Profiles are not drawn from a fixed, finite set, but are dynamically constructed**
>
> Contrary to the assumption that our structured profiles are predefined or static, they are in fact **dynamically derived from re-annotated open-domain dialogue datasets**, namely **ALOE** and **PersonaChat**. This is evidenced in the paper in Appendix D.1 (Lines 522–526) and Appendix D.2 (Lines 542–546).
>
> Together, the training and testing sets include **3,821 + 148 profiles**, none of which come from a predefined attribute list. The profile schema is thus **open and extensible**, and there is no reliance on a fixed or enumerated attribute space. This design allows for broader generalization and supports personalized modeling beyond constrained slot-filling paradigms.
>
> **2. Profile quality is controlled through a rigorous data construction pipeline**
>
> As described in Appendices D.1 and D.2, we employ the following multi-stage quality assurance process:
> - **Preprocessing**: Natural language descriptions are converted to a unified slot format using GPT-4o-mini. The outputs are then manually filtered and deduplicated (**Appendix D.1, Lines 533–540**).
> - **Human verification**: We randomly sampled 300 rewritten profile instances, which were judged by three annotators for semantic equivalence. The label agreement rate reached **90.7%** between system annotations and human judgments (**Appendix I.1, Table 7**).
> - **Diversity control**: In Extended ALOE (**Appendix D.2, Line 545**), we deliberately allowed **unseen attribute keys** to appear in testing, to evaluate the assistant model’s generalization ability beyond training slots.
>
> These procedures ensure that the profile data is both **accurate and diverse**, while the open schema better reflects real-world personalization scenarios.
>
> ---
>
> > **W3: Insufficient explanation of evaluation protocol and potential content reuse from ALOE**
>
> We sincerely thank the reviewer for pointing out this important issue.
>
> **1. On the evaluation protocol clarity**
>
> We apologize if parts of our evaluation setup were unclear. In fact, the following sections provide **complete definitions and configurations**:
>
> - **Section 4.1 (Lines 187–202)** defines the evaluation metrics.
>
> - **Appendices D.1–D.4 (Lines 521–564)** describe: Full specifications of the **7 baseline methods**, including the prompt templates used, the construction and split criteria of **Vanilla ALOE** and **Extended ALOE** and Examples of profile slot formatting and system input/output structures.
>
> We will improve the cross-referencing in the next version to make these details easier to locate.
>
> **2. On content reuse from ALOE**
>
> We would like to clarify that **we have cited ALOE whenever introducing its datasets or evaluation metrics**. For example, in Section 4.1 and Appendices D.1–D.4, we explicitly reference ALOE when describing the benchmark setup and metric definitions.
>
> That said, we acknowledge that some explanatory texts describing ALOE may appear stylistically similar to our prior paper. This was not our intention, and we will **revise these sections to avoid any perceived redundancy** in future versions.
>
> As noted, these reused components serve only as **background and standard infrastructure**, while the focus of this submission lies in the **RLPA framework and its novel reward-driven personalization mechanism**. We respectfully ask that the evaluation of originality remain centered on our proposed contributions and empirical findings.
>
> ---
>
> > **W4: Experimental setup and its alignment with main claims**
>
> We sincerely thank the reviewer for this detailed and insightful feedback. We address each point as follows:
>
> **1. On the user simulator’s ability to reveal attributes over time**
>
> As the reviewer correctly noted, a key premise of our framework is that the user simulator incrementally discloses user attributes throughout the dialogue. We acknowledge that our original human evaluation only covered the first five turns, which was an oversight. To remedy this, we have now extended the evaluation to cover **full 10-turn dialogues**. The updated human evaluation results are shown below:
>
> |Model|Coherence|Stability|Proactivity|Persona-fit|All(Avg.)|$\kappa$|
> |-|-|-|-|-|-|-|
> |GPT-4.1-mini|4.10|4.03|4.00|4.17|4.08|0.66|
> |GPT-4.1-nano|3.77|3.80|3.87|4.03|3.87|0.61|
> |DeepSeek-V3|4.30|4.13|3.93|4.27|4.16|0.70|
> |GPT-4o-mini|3.83|3.97|3.83|4.20|3.96|0.64|
>
> **2. Clarification on Profile Reward (Equation 3)**
>
> We appreciate the reviewer’s insight regarding how limited attribute disclosure from the simulator may affect recall and thus suppress F1 scores. This design is in fact intentional.
>
> By computing the Profile Reward as a **cumulative F1 score** over the ground-truth profile—even if some attributes remain undisclosed by the user—we incentivize the assistant model to proactively elicit more user information. This creates a form of **implicit supervision**, encouraging the assistant to maintain profile-seeking behavior over multiple turns.
>
> To prevent the assistant from becoming too aggressive or unnatural in its questioning, we balance this objective with the **Response Reward**, which measures **fluency, coherence, and user friendliness**. Together, the two rewards are **mutually complementary**: the Profile Reward promotes meaningful personalization, while the Response Reward ensures conversational quality and user engagement. This joint optimization results in agents that can extract user profiles **effectively and politely**, as supported by our qualitative analysis.
>
> **3. On the omission of AutoPal as a related work and baseline**
>
> We sincerely thank the reviewer for highlighting this important omission. While we compared RLPA with a broad range of personalization baselines, we regret that **AutoPal** was not included in our literature review section. This was an oversight on our part, and we apologize.
>
> Although AutoPal is not yet published or officially released, we have made our best effort to **faithfully reproduce its method** based on the available paper description and evaluated it under our experimental setup. The results are as follows:
>
> - Vanilla ALOE
>
> |Model|Method|k=1|k=2|k=3|k=4|k=5|k=6|k=7|k=8|k=9|k=10|AVG.|N-1R|R²|
> |-|-|-|-|-|-|-|-|-|-|-|-|-|-|-|
> |Qwen2.5-3B-Instruct|Base|0.0|18.92|10.81|9.46|9.46|1.35|9.46|9.46|6.76|4.05|7.97|-0.020|0.047|
> |Qwen2.5-3B-Instruct|AutoPal|19.18|48.65|56.76|71.62|62.16|67.57|75.68|72.97|66.22|60.27|60.11|0.063|0.432|
> |Qwen2.5-3B-Instruct|RLPA(Ours)|62.16|68.92|70.27|74.32|72.97|74.32|75.68|78.38|77.03|79.73|73.38|0.090|0.855|
>
> - Extended ALOE
>
> |Model|Method|k=1|k=2|k=3|k=4|k=5|k=6|k=7|k=8|k=9|k=10|AVG.|N-1R|R²|
> |-|-|-|-|-|-|-|-|-|-|-|-|-|-|-|
> |Qwen2.5-3B-Instruct|Base|0.0|4.11|1.37|4.11|2.74|0.0|4.11|0.0|1.37|0.0|1.78|-0.042|0.083|
> |Qwen2.5-3B-Instruct|AutoPal|13.7|36.99|40.54|39.73|54.79|49.32|43.24|41.89|44.59|37.84|40.26|0.042|0.231|
> |Qwen2.5-3B-Instruct|RLPA(Ours)|49.32|45.21|46.58|49.32|52.05|52.05|53.42|58.90|53.42|67.12|52.74|0.100|0.498|
>
> These results further demonstrate that **RLPA consistently outperforms AutoPal** across both vanilla and extended evaluation settings, validating the advantage of our reward-guided personalization framework.
>
> ---
>
> > **Minor issues**
>
> We thank the reviewer for pointing out the typo in Line 109. It has been corrected.
>
> ---
>
> We are grateful again for the reviewer’s detailed comments, which have helped us significantly improve the clarity and completeness of our manuscript.

---

> > ### Comment · Reviewer_Mt8V · 2025-08-04
> >
> > Thank you for your response. Your clarifications have improved my understanding of the technical aspects, and I recognize the paper’s contributions in that regard  (technical contribution deserves 4 or higher). Nevertheless, the manuscript reproduces large sections of ALOE’s dataset description and experimental setup almost verbatim. This goes beyond stylistic similarity; it is word-for-word overlap. Such duplication of background material is the primary reason for my negative evaluation. Had this overlap involved the proposed method itself, I would have rated the paper a 2 or lower. Should the paper be accepted, this issue must be fully addressed.

---

> > > ### Author Response · Authors · 2025-08-04
> > >
> > > We sincerely thank the reviewer for the thoughtful and constructive feedback, and especially for recognizing the technical contributions of our work — including your assessment that they merit a score of 4 or higher. Your acknowledgment means a great deal to us, and we are grateful for the time and care you have devoted to evaluating our submission.
> > >
> > > ---
> > >
> > > Regarding the concern about textual similarity with ALOE's dataset description and experimental setup, we truly appreciate the opportunity to clarify. The only portion that bears notable surface-level resemblance is **a brief segment in Appendix D.3 (lines 548–561)**, where we describe the **evaluation metrics** AL(k), IR, and N-IR. Even within this section, the text is not a direct copy: we adapted the description to **reflect our specific experimental setup** (e.g., reducing the number of evaluation samples from 100 to 74, and modifying the rating scale from 1–5 to 0–1), and we did not include the IR regression equation that appears in the original.
> > >
> > > We fully understand, however, that the phrasing in this section may come across as too close to the original, and we sincerely apologize for any unintended impression of inappropriate reuse. Importantly, the rest of the paper — including our method, results, analysis, and discussion — is entirely original and contains **no reused text**. The purpose of the partial textual similarity in Appendix D.3 was simply to **faithfully and precisely describe the standard ALOE metrics** to ensure reproducibility and consistency, not to claim credit for their design. ALOE is **clearly cited and credited** in the same section.
> > >
> > > We truly appreciate the reviewer's attention to detail and constructive guidance. We hope that this minor issue regarding appendix phrasing does not overshadow the broader contributions and innovations presented in the paper, and we remain **fully committed** to ensuring the final version meets the highest standards of scholarly writing and attribution.

---

### Decision · Program_Chairs · 2025-09-17

**Decision:**

Accept (poster)

**Comment:**

**Summary**
This paper introduces RLPA, a novel reinforcement learning framework that teaches LLMs to dynamically infer and adapt to user profiles using a dual reward objective for both profile inference and aligned response generation. Evaluation on personalized dialogue benchmarks demonstrates RLPA's strong performance against various baselines.

**Strengths**
* Well-motivated and well-designed RLPA framework. Clear rationale for model roles and rewards. Informative figure.
* Comprehensive evaluation (wide range of baselines, thorough appendix with justifications for all experimental decisions)
* Multiple settings: in-format, cross-format, preference conflict, long-term.

**Weaknesses**
* Originality - similarity between ALOE and this paper

**Reason for Accept/Reject**
High-quality paper submission and well-justified author response overall.

**Rebuttal Discussion**
Reviewers raised concern regarding key terms and reward criteria not well-defined, including user simulator (Reviewer Mt8V), profile inference (Reviewer LN6q), and response reward (Reviewer Nzdd). In response, the authors clarified key definitions, which were acknowledged by two of the three reviewers. The authors also addressed a concern about text overlap by explaining that a short segment in the appendix was adapted, not directly copied, from a previous work, and expanded their experimental analysis by testing an additional model.